# Differentiation of IL-26$^+$ T$_H$17 intermediates into IL-17A producers via epithelial crosstalk in psoriasis

Anissa Fries[1], Fanny Saidoune[1], François Kuonen[1], Isabelle Dupanloup[2], Nadine Fournier[2], Ana Cristina Guerra de Souza[2], Muzlifah Haniffa[3], Feiyang Ma[4], Johann E. Gudjonsson[4], Lennart Roesner[5,6], Yang Li[7], Thomas Werfel[5,6], Curdin Conrad[1], Raphael Gottardo[8], Robert L. Modlin[9], Jeremy Di Domizio[1]✉ & Michel Gilliet[1]✉

Interleukin (IL)-26 is a T$_H$17 cytokine with known antimicrobial and pro-inflammatory functions. However, the precise role of IL-26 in the context of pathogenic T$_H$17 responses is unknown. Here we identify a population of blood T$_H$17 intermediates that produce high levels of IL-26 and differentiate into IL-17A-producing T$_H$17 cells upon TGF-β1 exposure. By combining single cell RNA sequencing, TCR sequencing and spatial transcriptomics we show that this process occurs in psoriatic skin. In fact, IL-26+ T$_H$17 intermediates infiltrating psoriatic skin induce TGF-β1 expression in basal keratinocytes and thereby promote their own differentiation into IL-17A-producing cells. Thus, our study identifies IL-26-producing cells as an early differentiation stage of T$_H$17 cells that infiltrates psoriatic skin and controls its own maturation into IL17A-producing T$_H$17 cells, via epithelial crosstalk involving paracrine production of TGF-β1.

T$_H$17 cells belong to an effector T cell subset defined by the ability to produce interleukin (IL)-17 family cytokines including IL-17A and its homologs[1,2]. T$_H$17 cells are critically involved in immune responses against fungi and extracellular bacteria, but also contribute to the pathogenesis of autoimmune diseases such as psoriasis, rheumatoid arthritis, inflammatory bowel disease and multiple sclerosis[3]. In psoriasis, IL-17A activates keratinocytes to release neutrophil-attracting chemokines and antimicrobial peptides[4], thereby critically contributing to the disease pathogenesis. IL-26 is another cytokine produced by pathogenic T$_H$17 cells. IL-26 can kill bacteria and drive innate activation

of dendritic cells (DC) by forming complexes with DNA[5,6]. IL-26 can also activate keratinocytes via the IL-26-receptor (IL-26-R)[7] and induce secretion of IL-8 (CXCL8), IL-1β, chemokine (C-C motif) ligand 20, and IL-33[8]. Despite these described functions, the precise role of IL-26 in the context of pathogenic IL-17A responses is still unclear.

The differentiation of T$_H$17 cells into IL-17A-producing cells requires both inflammatory cytokines (IL-6, IL-1β, and TNF) and TGF-β[9–11]. The TGF-β-dependent induction of IL-17A producing cells involves several mechanisms[12], but the inhibition of SOCS3 with enhancement of STAT3, which regulates the transcription of the

[1]Department of Dermatology, CHUV University Hospital and University of Lausanne (UNIL), Lausanne, Switzerland. [2]Translational Data Science Facility, Agora Cancer Research Center, Swiss Institute of Bioinformatics, Lausanne, Switzerland. [3]Department of Dermatology and NIHR Newcastle Biomedical Research Centre, Newcastle Hospitals NHS Foundation Trust, Newcastle upon Tyne NE2 4LP, UK. [4]Department of Dermatology, University of Michigan, Ann Arbor, MI 48109, USA. [5]Department of Dermatology, Hannover Medical School, Hannover, Germany. [6]Cluster of Excellence RESIST (EXC 2155), Hannover Medical School, Hannover, Germany. [7]Department of Computational Biology for Individualised Medicine, Centre for Individualised Infection Medicine (CiiM), Helmholtz Centre for Infection Research (HZI), Hannover Medical School (MHH), Hannover, Germany. [8]Biomedical Data Sciences Center, CHUV, UNIL, and SIB, Lausanne, Switzerland. [9]Division of Dermatology, Department of Medicine, University of California, Los Angeles, CA 90095, USA. ✉e-mail: Jeremy.Di-Domizio@chuv.ch; michel.gilliet@chuv.ch

lineage-specific transcription factor retinoic-acid-receptor-related orphan nuclear receptor t (RORγt) and IL-17 genes[3,13], appears to be a key one. TGF-β drives pathogenic T$_H$17 responses in autoimmune diseases such as psoriasis. In fact, increased epidermal expression of TGF-β1 in psoriasis is associated with disease activity[14–16] and TGF-β1 gene polymorphisms represent risk factors for psoriasis development[17,18]. Furthermore, mice with epidermal TGF-β1 over-expression develop severe IL-17A-dependent psoriasis-like skin disease[19,20]. However, the mechanisms that drive TGF-β expression in psoriasis and how they are linked to pathogenic IL-17A responses[21] remain unclear.

Here, we uncover the link between IL-26, TGF- β1 and pathogenic IL-17A producing cells. We show IL-26$^+$ T$_H$17 represent an early T$_H$17 differentiation stage, which accumulate in psoriatic skin lesions and produce IL-26, which triggers TGF-β1 expression in keratinocytes. TGF-β1 then conditions infiltrating IL-26$^+$ T$_H$17 to mature into IL-17A-producing cells, suggesting a unique tissue control of pathogenic T $_H$17 cells in psoriasis via epithelial crosstalk.

## Results

### T$_H$17 cells contain high numbers of IL26$^+$ T cells that do not express IL-17A

To gain insights into the role of IL-26 in pathogenic T$_H$17 cell responses, we compared the expression of IL-26 and IL-17A in bona-fide blood T$_H$17 cells, isolated from circulating CD45RO+ T cells of healthy donors based on the expression of CCR6 and CCR4 and the absence of CXCR3. Because we were unable to obtain reproducible data on intracellular IL-26 protein staining, we performed mRNA staining using fluorescent hybridization probes followed by flow cytometry. We found that blood T$_H$17 cells contained a large population of *IL26*-expressing cells that does not express *IL17A* (Fig. 1a, b). These IL26$^+$IL17A$^-$ T cells were significantly more abundant than IL17A$^+$ T cells, including both IL17A$^+$IL26$^+$ and IL17A$^+$IL26$^-$ T cells (Fig. 1a, b). IL26$^+$IL17A$^-$ T cells appeared to exclusively express IL-26, as similar populations of IL-26 single-positive T cells were detected among blood T$_H$17 cells upon co-staining for other cytokines including IFN-γ, IL-22, IL-21, IL-13, IL-10 and IL-9 (Fig. 1a, Supplementary Fig. 1). We also generated stable clones from purified T$_H$17 cells and obtained 32 clones, of which 21 were IL26$^+$IL17A$^-$ clones and 11 were IL26$^+$IL17A$^+$ double positive. Figure 1c shows four representative clones, two IL26$^+$IL17A$^-$ and two IL26$^+$IL17A$^+$ double positive ones. No IL17A+ single-positive clones were obtained. All clones contained high numbers of IL26$^+$ T cells that were significantly more abundant than IL17A$^+$ T cells, similar to the observation made in primary T$_H$17 cells (Fig. 1c, d). IL26$^+$ T cells within the T$_H$17 clones co-expressed some levels of *IL22* and variable amounts of *IFNG* (Fig. 1c). These results indicate that blood T$_H$17 cells contain an abundant population of IL26-single expressing T cells that is distinct from IL17A-expressing T cells.

### IL26$^+$ T cells develop early during T$_H$17 cell differentiation, whereas IL17A$^+$ T cells appear later and require TGF-β1

To understand the factors that drive the generation of IL26$^+$ T cells, we generated T$_H$17 cells from naive T cells stimulated with anti-CD3 and CD28 antibodies in the presence of the standard T$_H$17 polarizing cocktail that combines IL-1β, IL-6, IL-23 and TGF-β[22,10,23,9]. We also simulated naive T cells in the presence of the T$_H$17 cocktail without TGF-β1 or with the single cytokines alone (IL-1β, IL-6, or IL-23) to assess the requirement of each cytokine. Within 7 days of culture, the full T$_H$17 cocktail promoted the differentiation of high numbers of IL26$^+$IL17A$^-$ T cells and moderate numbers of IL17A-expressing T cells, which included both IL17A$^+$IL26$^+$ and IL17A$^+$IL26$^-$ T cells (Fig. 2a, b). The single cytokines (IL-1β, IL-6, or IL-23) were sufficient to induce the generation of IL26$^+$IL17A$^-$ T cells and their combination (IL-1β, IL-6, plus IL-23) induced even greater numbers of these cells. By contrast, this cocktail only induced very low numbers of IL17A$^+$ T cells unless TGF-β1

was added (Fig. 2a, b). We confirmed these data at the protein level showing that IL-1β, IL-6, or IL-23 alone or in combination strongly induced IL-26 secretion, whereas IL-17A was barely detectable in the supernatants and required the addition of TGF-β1 for its induction (Fig. 2c).

Time-course analysis of T$_H$17 differentiation revealed *IL26* expression as early as 3 days after activation of naive T cells, with a peak after 5 days, regardless of the presence of TGF-β1 in the cultures (Fig. 2d, e). By contrast, *IL17A* expression was only detectable after 5 days in TGF-β1 supplemented cultures, reaching maximal levels at day 7 (Fig. 2d, e). Together these results indicate that the priming requirements and the timing of IL-26$^+$ and IL-17A$^+$ T cell development within T$_H$17 cells are distinct: IL-26$^+$ T cells develop early and require few cytokines, whereas IL-17A$^+$ T cells arise later during T$_H$17 differentiation and require the presence of TGF-β1.

### IL26$^+$ T cells are T$_H$17 cell intermediates that can differentiate into IL17A$^+$ T cells upon TGF-β1 exposure

Having found that TGF-β1 induces IL-17A at late stages of T$_H$17 differentiation, we asked whether IL-26$^+$ T cells, which develop early, could switch to IL-17A expression in the presence of TGF-β1. To address this question, we stimulated purified blood T$_H$17 cells, which contain high numbers of IL-26$^+$ T cells, with anti-CD3/CD28 in the presence of TGF-β1. Before culture, purified memory T$_H$17 cells indeed contained high numbers of IL26$^+$IL17A$^-$ T cells and less IL17A$^+$ T cells (Fig. 3a, b). Upon TGF-β1 exposure, we observed increasing percentages of IL17A$^+$IL26$^+$ double-positive cells within T$_H$17 cells, gradually developing from IL26$^+$IL17A$^-$ single-positive cells (between day 3 and day 5 of culture) (Fig. 3a, b). Subsequently (day 7), the IL17A$^+$IL26$^+$ double-positive cell population progressively lost *IL26* expression, leading to an increase of IL17A$^+$IL26$^-$ single-positive cells (Fig. 3a, b). At day 7, T$_H$17 cell had switched their phenotype from IL26$^+$ cells as being most abundant to a predominance of IL17A$^+$ cells (Fig. 3a, b), a finding that was also confirmed on the protein level (Fig. 3c). Altogether, our results indicate that IL-26$^+$ cells may represent early T$_H$17 cell differentiation intermediates that can further mature into IL-17A-producing T cells in the presence of TGF-β1.

### IL26$^+$ T$_H$17 intermediates infiltrating psoriatic skin differentiate into TGF-β imprinted IL17A$^+$ T cells

Because psoriasis is associated with increased lesional TGF-β and IL-17A expression, we next sought to determine whether IL17A+ producing T cells in psoriatic skin differentiate from IL26$^+$ T$_H$17 intermediates and whether this process is driven by TGF-β. Single-cell RNAseq data generated from psoriasis skin lesions[24] revealed that dermal CD4 T cells in nonlesional skin contained IL26$^+$ T cells (green) only, whereas dermal CD4 T cells in the lesions contained both IL26$^+$ (green) and IL17A$^+$ T cells (red) (Fig. 4a, b). These IL17A$^+$ T cells comprised both IL17A$^+$IL26$^+$ double-positive T cells and IL17A$^+$IL26$^-$ T cells (Fig. 4c), potentially reflecting the stepwise acquisition of *IL17A* and loss of *IL26* observed in vitro. To confirm this progression in-vivo, we performed pseudotime analysis of the scRNAseq data to order psoriatic CD4 T cells along temporal trajectories based on their gene expression profiles (Fig. 4d–f). Within these temporal trajectories, CCR7, used as a marker for undifferentiated T cells, was depicted on T cells at early timepoints and lost thereafter, consistent with T cell differentiation into effector cells (Fig. 4e). *IL26* expression was also depicted at early timepoints of pseudotime analysis, whereas *IL17A* appeared later on the same trajectories (Fig. 4d, g). These data are consistent with IL-26$^+$ T cells being an early differentiation intermediate of T$_H$17 cells and demonstrates that differentiation into IL17A$^+$ T cells occurs within psoriatic skin. Interestingly, we observed that IL26$^+$ T cells of non-lesional skin projected onto the same trajectories at even earlier timepoint (pseudotime <3) compared to their lesional counterparts

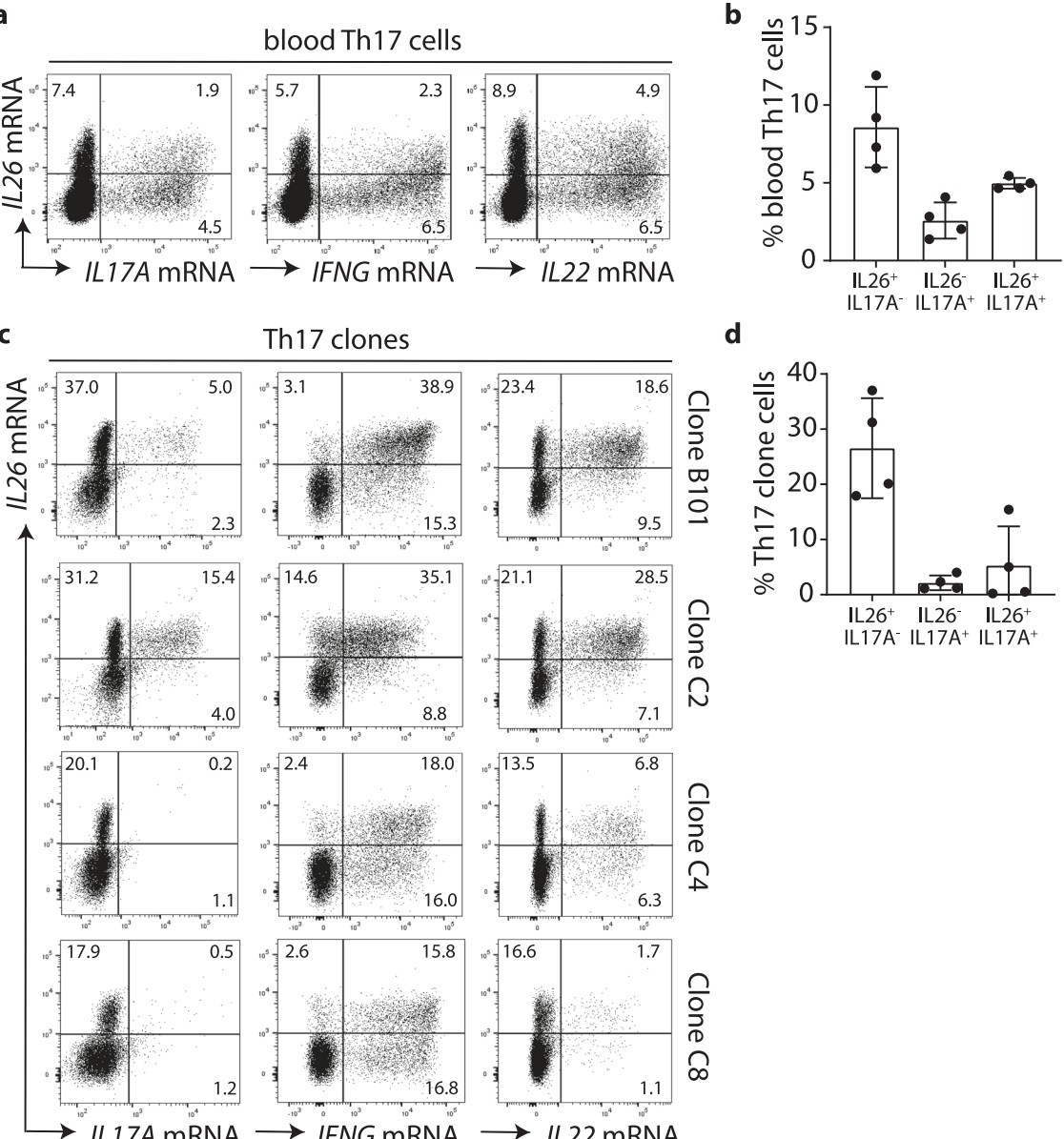

**Fig. 1 | Circulating T$_H$17 cells contain a large cell population expressing IL26 but not IL17A. a** Flow-FISH analysis of IL-26, IL-17A, IFN-γ, and IL-22 mRNA expression in memory T$_H$17 cells isolated from peripheral blood. Data shown are from one representative donor. **b** Frequencies of IL26 and IL17A single producing T cells and IL26/IL17A double producers among blood memory T$_H$17 cells. Data represent the mean ± SD (*n* = 4 independent donors). **c** Flow-FISH analysis of IL-26, IL-17A, IFNγ, and IL-22 mRNA expression in four independent T$_H$17 cell clones (B101, C2, C4, and C8). **d** Frequencies of IL26 and IL17A single-positive cells, and IL26/IL17A double positive cells among T$_H$17 cell clones Data are mean ± SD (*n* = 4). Source data are provided as a Source Data file.

(Supplementary Fig. 2A, B), suggesting a differentiation continuum of T$_H$17 cells starting from IL26$^+$ T$_H$17 intermediates in nonlesional skin all the way to IL17A$^+$ T cells in psoriatic skin lesions. Strikingly, the majority of IL26$^+$ T cells in both nonlesional and lesional skin of psoriasis patients as well as in healthy skin expressed the TRM marker CD103 (*ITGAE*) (Supplementary Fig. 2C), indicating that IL26$^+$ T$_H$17 intermediates in the skin derive from skin-resident rather than circulating T cells.

Confirming our in vitro data, IL26$^+$ T cells depicted in psoriatic skin at early pseudotime were IL26-single producers with no expression of IL-17 cytokines IL17A and IL17F, nor other T$_H$17-associated cytokines such as IL-22, IL-21, and GM-CSF (Supplementary Fig. 2D, E). At intermediate pseudotime we observed the appearance of IL17A$^+$ T cells as equal numbers of IL17A$^+$IL-26$^+$ and IL17A$^+$IL26$^-$ T cells (Supplementary Figs. 2D and 1E). At late pseudotime we observed a predominance of IL17A$^+$IL26$^-$ T cells (73%, Supplementary Fig. 2C, D),

confirming the gradual shift from IL26$^+$ to IL17A$^+$ T cells in psoriatic skin lesions. Other T$_H$1- and T$_H$17-associated cytokines including IL-22, IL-21, IFN-γ, TNF, and GM-CSF appeared at intermediate to late timepoints, whereas T$_H$2-associated cytokines (IL-4, IL-9, and IL-13) remained undetectable (Supplementary Fig. 2C, D).

To demonstrate conclusively that IL-17A producing T cells arise from IL26$^+$ T cells, we followed temporal trajectories of TCR clonotypes in psoriatic skin by using a dataset that integrates scRNAseq with scTCRseq[25]. As in the previous dataset, we confirmed temporal CD4 T cell trajectories from early/intermediate stages containing higher numbers of *IL26* expressing cells to late stages containing the majority of IL17A-expressing cells (Fig. 5a, b). Among the 646 CD4 T cell clones identified, 70 clones (19%) appeared to be present across the three temporal stages pointing to a differentiation process (Fig. 5c). When specifically looking at clones expressing *IL17A* at late stages (*n* = 9), 4/9 of clones had their IL26$^+$ T cell correspondence at early

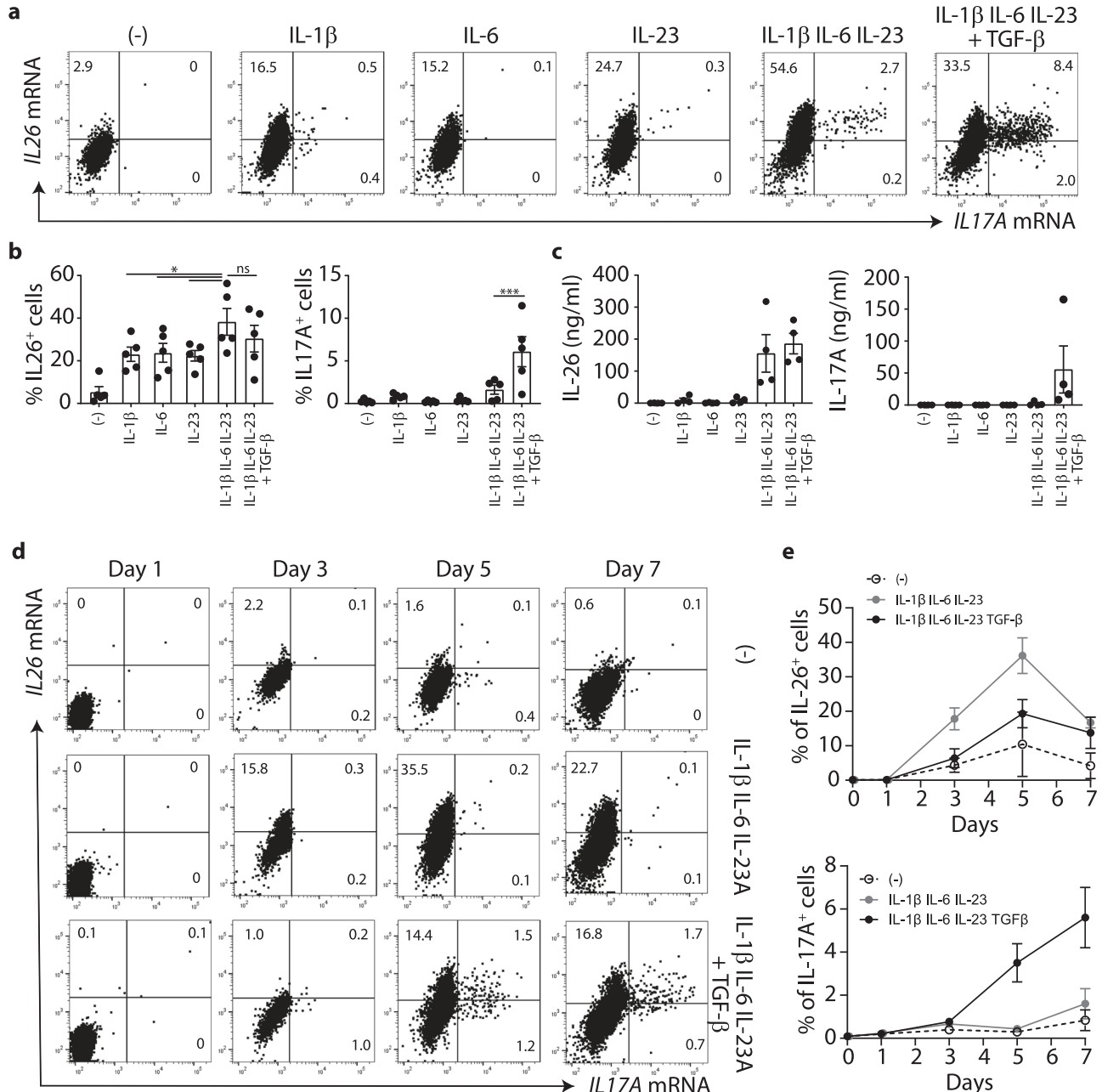

**Fig. 2 | Rapid TGF-β1 independent differentiation of IL26⁺ but not IL17A⁺ T_H17 cells. a** Flow-FISH analysis of IL-26 and IL-17A mRNA expression in naive CD4 T cells stimulated for 7 days in the presence of either single cytokines or a mix of the T_H17 polarizing cytokines IL-1β, IL-6, IL-23, with or without TGF-β1. Data from one representative donor are shown. **b** Percentages of IL26⁺ and IL17A⁺ T cells in cultures described in (**a**). Data are mean + SEM of 5 independent donors. Data were statistically analyzed using one-way ANOVA followed by Tukey's multiple comparisons test. *$p = 0.0255$; ***$p = 0.0003$. **c** ELISA measurement of secreted IL-26 and IL−17A by T cells obtained by a 7-day stimulation of naive CD4 T cells as in (a). Data are mean ± SEM of 4 independent donors. **d** Flow-FISH analysis of the expression kinetics of IL-26 and IL-17A mRNA in naive CD4 T cells stimulated for 7 days either alone, or in the presence of T_H17 polarizing cytokines IL-1β, IL-6, IL-23, with or without TGF-β1. Data from one representative donor are shown. **e** Percentages of IL26⁺ and IL17A⁺ T cells cultured as in (**d**). Data are mean ± SD of 3 independent donors. Source data are provided as a Source Data file.

stages (Fig. 5d, e). These data confirm, at the clonal level, that IL26⁺ T_H17 intermediates differentiate into IL17A⁺ T cells.

To determine whether the maturation of IL26⁺ into IL17A-producing T_H17 cells in psoriatic skin occurs via TGF-β, we first investigated *TGFBR1* and *TGFBR2* expression on IL26⁺ and IL17A⁺ CD4⁺ T cells. We found that *TGFBR1* and *TGBR2* were highly expressed in IL26⁺ T cells but decreased in IL17A⁺ T cells (Supplementary Fig. 3A), suggesting responsiveness to TGF-β by IL26⁺ T_H17 intermediates and a potential decrease in receptor expression in IL17A⁺ T cells following TGF-β-mediated activation. In fact, using a unique TGF-β-induced

transcriptional signature recently described in the skin (TGF-β signature genes include *NR4A1, NR4A2, HSPA1A, HSPA1B, HSPA6, JUNB, SOCS3, DUSP6, CISH, CTLA4*, and *MALAT1*)[26], we found that expression of these TGF-β signature genes was increased along the pseudotime and associated with the expression of IL-17A but not IL-26 (Supplementary Fig. 3B–D). Thus, T_H17 cells in psoriatic skin transition from IL26⁺ T_H17 intermediates to IL-17A producers under the influence of TGF-β. To investigate whether this transition specifically occurs in psoriatic skin, we turned to our atopic dermatitis (AD) scRNAseq dataset to quantify the number of IL26⁺ and IL17A⁺ cells. Interestingly,

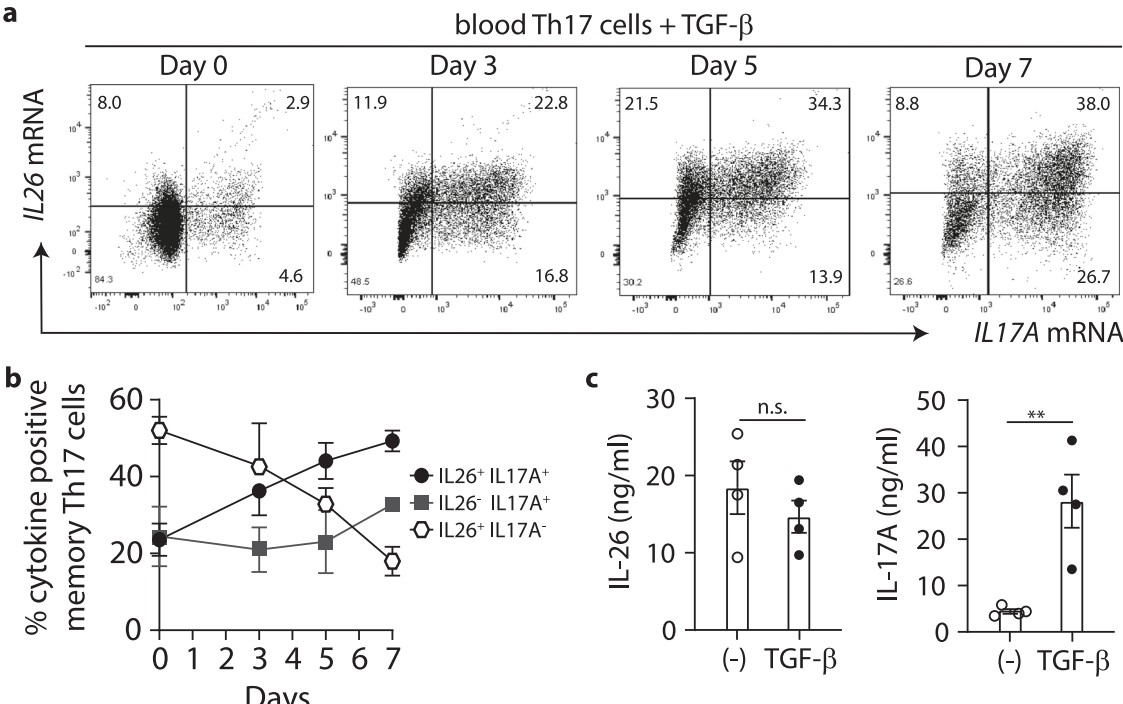

**Fig. 3 | IL26$^+$ T$_H$17 cells differentiate into IL-17A producers in the presence of TGF-β1. a** Flow-FISH analysis of IL-26 and IL-17A mRNA expression in blood memory T$_H$17 cells re-stimulated with anti-CD3/CD28 for 7 days in the presence of TGF-β1. Data from one representative donor is shown. **b** Percentages of IL26 and IL17A single-positive cells, and IL26/IL17A double positive cells among blood T$_H$17 cells stimulated as in (**a**). Data are mean ± SD of 3 independent donors. **c** ELISA measurement of IL-26 and IL-17A secretion by T$_H$17 cells stimulated for 7 days with or without addition of TGF-β1. Data are mean ± SEM of 4 independent donors. Data were statistically analyzed using two-tailed unpaired Student's t-test. **p = 0.0065. Source data are provided as a Source Data file.

high numbers of IL26$^+$ cells were also found in AD lesional skin, similar to psoriasis (Fig. 5f). However, when comparing the numbers of IL17A$^+$ cells, we found only few IL17A$^+$ cells in AD compared to psoriasis (Fig. 5g). Despite the lower number, we found pseudotime trajectories between IL26$^+$ and these IL17A$^+$ T cells (Fig. 5h), indicating that the skin transition from IL26$^+$ T$_H$17 intermediates to IL-17A producers is a general principle that preferentially occurs in psoriasis.

### IL26$^+$ T$_H$17 intermediates trigger TGF-β1 production by keratinocytes

Having demonstrated that TGF-β drives the late differentiation of IL26$^+$ T$_H$17 intermediates into IL-17A-producing T cells and having found a TGF-β gene signature in psoriatic IL-17A-producing T cells, we next sought to investigate the mechanism that drive TGF-β expression in psoriatic skin. Psoriatic skin lesions expressed elevated levels of *TGFB1* tissue mRNA compared to nonlesional skin or healthy skin (Fig. 6a and Supplementary Fig. 4A). Single-cell RNAseq analysis revealed that in psoriatic skin lesion, *TGFB1* was strongly expressed in a subset of keratinocytes that mapped to the same UMAP location in all three patients analyzed (Fig. 6b). By contrast, analysis of nonlesional skin did not yield strong expression of *TGFB1*, consistent with the lack of IL-17A producers (Fig. 6b). RNA in situ hybridization (RNA ISH) and immunofluorescence analysis of psoriatic skin showed that TGF-β1 was expressed by basal and suprabasal keratinocytes in the lower epidermal layers (Fig. 6c and Supplementary Fig. 4B). scRNAseq data confirmed expression of *TGFB1* in basal and proliferating psoriatic keratinocytes mapping to the lower epidermis but not in the more differentiated suprabasal keratinocyte population (Fig. 6d). When compared to psoriasis, *TGFB1* expression in keratinocytes was markedly lower or even absent in AD (Fig. 6d).

Then, to investigate the mechanism of TGF-β induction, we stimulated healthy keratinocytes (NHEK) in vitro with a number of cytokines characterizing the psoriasis microenvironment including

IL-1β, IL-6, IL-17A, IL-19, IL-22, IL-23, IL-26, and IFN-γ. Strikingly, only stimulation with IL-26 but not with the other cytokines triggered high levels of *TGFB1* in keratinocytes (Fig. 6e). IL-26 also induced the expression of *TGFB1* in ex-vivo cultures of healthy skin explants (Fig. 6f). Because the IL-26 receptor (IL-26R) is a heterodimer composed of the IL-10R2 and IL-20R1, a combination that is exclusively found in keratinocytes from both healthy and psoriatic skin (Supplementary Fig. 5A, B), we next asked whether keratinocyte activation by IL-26 occurs through IL-26R. In fact, the IL-26-driven *TGFB1* expression was completely inhibited by neutralizing antibodies against both IL-10R2 and IL-20R (Fig. 6g), indicating an IL-26R-mediated activation of keratinocytes. These data were also confirmed by CRISPR-Cas9-mediated genetic ablation of *IL20R1* in keratinocytes (Fig. 6h). Furthermore, supernatants from activated blood T$_H$17 cells induced TGF-β1 in keratinocytes via IL-26 as this induction was abrogated in the presence of anti-IL-10R2 and anti-IL-20R1 antibodies (Fig. 6i).

Finally, we sought to determine whether TGF-β production by IL-26-stimulated keratinocytes is sufficient to induce IL-17A-producing T cells by culturing purified T$_H$17 cells with supernatants of IL-26-stimulated keratinocytes with or without neutralizing antibodies against TGF-β. After 7 days, cultured T$_H$17 cells were restimulated before IL17A$^+$ and IL26$^+$ T cells were quantified by FISH-flow. Supernatant from IL-26-stimulated keratinocytes strongly enhanced the number of IL-17A-producing T cells and this was dependent on TGF-β, as it was blocked by neutralizing antibodies against TGF-β (Fig. 6j). Altogether, these results suggest that early T$_H$17 cell intermediates producing IL-26 stimulate the secretion of TGF-β by keratinocytes, which drives the induction of IL-17A-producing T cells in psoriatic skin.

### Psoriatic keratinocytes in the vicinity of infiltrating IL26$^+$ T$_H$17 intermediates express TGF-β1

Having demonstrated that IL-26 drives the TGF-β production by keratinocytes, we sought to determine whether this process also

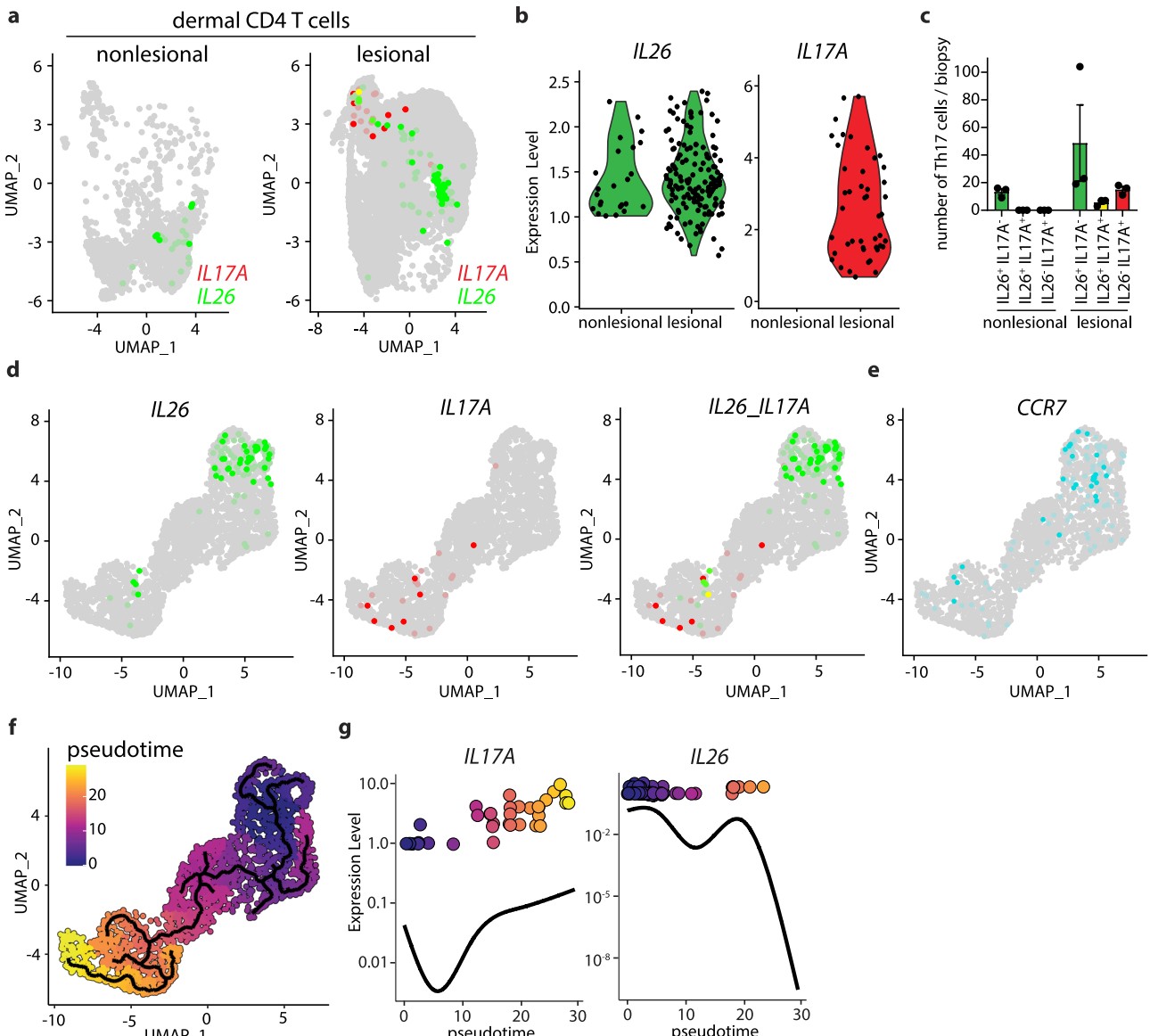

**Fig. 4 | Psoriatic skin contain IL26⁺ T cells that differentiate into TGF-β−imprinted IL17A⁺ T cells. a** UMAP projection of the single-cell transcriptomes of dermal CD4 T cells from nonlesional (left) and lesional (right) skin of 3 psoriasis patients colored according to the expression level of IL-26 (green) and IL-17A (red). **b** Expression level of IL-26 and IL-17A in dermal CD4 T cells as in (**a**). **c** Number of IL26 and IL17A single-positive, and IL26/IL17A double positive dermal CD4 T cells in nonlesional and lesional skin of psoriasis patients. Data represent the mean ± SEM of 3 independent patient samples. **d, e** UMAP projection of the single-cell transcriptomes of cytokine-producing dermal CD4 T cells from lesional skin colored according to the expression level of *IL26* (green), *IL17A* (red), and *CCR7* (blue). **f** UMAP projection of the single-cell transcriptomes of dermal CD4 T cells as in (a) colored according to the inferred pseudotime. Solid lines are trajectories of gene expression changes learned by Monocle 3. **g** Dynamics of *IL17A* and *IL26* gene expression in dermal CD4 T cells over pseudotime. Solid lines show the expression average. Source data are provided as a Source Data file.

occurs in psoriatic skin. Spatial transcriptomics analysis revealed that *TGFB1* expression by keratinocytes (Fig. 6b) mapped to lower layers of the psoriatic epidermis (purple dots, Fig. 7a). *IL26* expression by T cells (Supplementary Fig. 6A, B) was found at this same location (dots with white ring, Fig. 7a) with 97.4% of IL26-expressing spots at the dermo-epidermal junction also expressing *TGFB1* (Fig. 7b, c). By performing distance analysis between spots, we found that spots expressing T cell markers and positive for *IL26* were significantly closer to spots containing *TGFB1* than *IFNG*-positive spots (Fig. 7d, e), confirming that a spatial link to TGF-β1 expression by psoriatic keratinocytes and infiltrating IL26⁺ T cells. Confocal microscopy indeed confirmed that IL-26-producing T cells accumulate at the dermo-epidermal junction of psoriatic lesions, adjacent to TGF-β1-producing keratinocytes (Fig. 7f).

Previous studies have shown that IL-26 induces the expression of CXCL8, IL-1β, CCL22, and IL-33 in keratinocytes[8]. However, we observed that, in psoriatic skin, these factors were expressed at locations not associated with IL-26 expression. *CXCL8* was predominantly expressed by psoriatic keratinocytes of spinous layer (Supplementary Fig. 7A), while *IL1B, IL33*, and *CCL20* were principally expressed by dermal monocytes/macrophages and fibroblasts, and not co-localizing with IL26⁺ T cells (Supplementary Fig. 7B–D). Together these data indicate that TGF-β expression in psoriatic skin occurs in keratinocytes of the lower epidermis and suggest specific induction by IL26⁺ T cells. Having shown that AD contains similar high numbers of IL26⁺ T_H17 intermediates but lower expression levels of epidermal TGF-β1 and reduced numbers of IL17A⁺ cells when compared to psoriasis, we sought to compare the location of IL-26

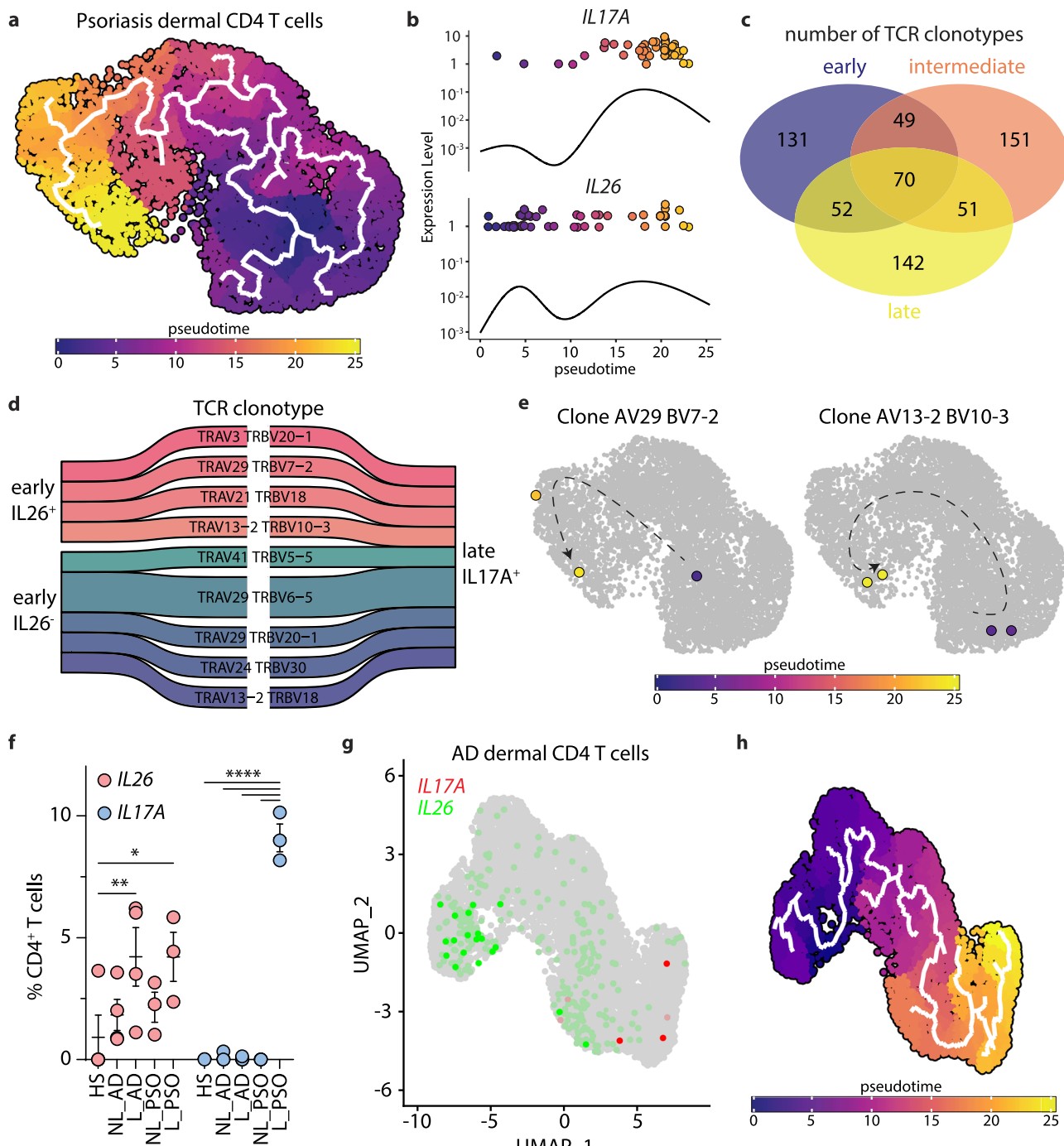

**Fig. 5 | IL26+ cells differentiate into IL-17A-producing cells at the clone level.**
**a** UMAP projection of the single-cell transcriptomes of nonlesional and lesional psoriatic skin CD4 T cells from two merged datasets colored according to the inferred pseudotime. **b** Dynamics of *IL17A* and *IL26* gene expression in skin CD4 T cells over pseudotime. Solid lines show the expression average. **c** Venn diagram of the number of shared TCR clonotypes between CD4 T cells at early (<10), inter-mediate, and late (>15) pseudotime. **d** Sankey diagram of the shared TCR clono-types between late IL17A⁺ CD4 T cells and early IL26⁺ or IL26⁻ CD4 T cells· **e** UMAP projection of the single-cell transcriptomes of two different TCR clones colored according to the inferred pseudotime. **f** Percentages of IL26⁺ and IL17A⁺ cells among CD4⁺ T cells from healthy (HS), nonlesional (NL), and lesional (L) skin of atopic dermatitis (AD) and psoriasis (PSO) patients. Data were statistically analyzed using two-way ANOVA followed by Tukey's multiple comparisons test. *$p = 0.0181$, **$p = 0.0094$, ****$p < 0.0001$. Data represent the mean ± SEM of 3 independent patient samples. **g** UMAP projection of the single-cell transcriptomes of dermal CD4 T cells from AD lesional skin colored according to the expression level of *IL26* (green), and *IL17A* (red). **h** UMAP projection of the single-cell transcriptomes of dermal CD4 T cells as in (**g**) colored according to the inferred pseudotime. Solid lines are trajectories of gene expression changes learned by Monocle 3. Source data are provided as a Source Data file.

expression in both diseases using spatial transcriptomics. We found similar IL26+ spots in the dermis of AD and psoriasis, but high numbers of epidermal IL26+ spots only in psoriasis (Fig. 7g). This finding was paralleled by the presence of high numbers of IL17A⁺ spots with epidermal contact only in psoriasis but not in AD (Fig. 7g), suggesting that the efficient transition of IL26+ TH17 intermediates into IL-17A producers only occurs in psoriasis due to the epidermal infiltration of the IL26+ TH17 intermediates.

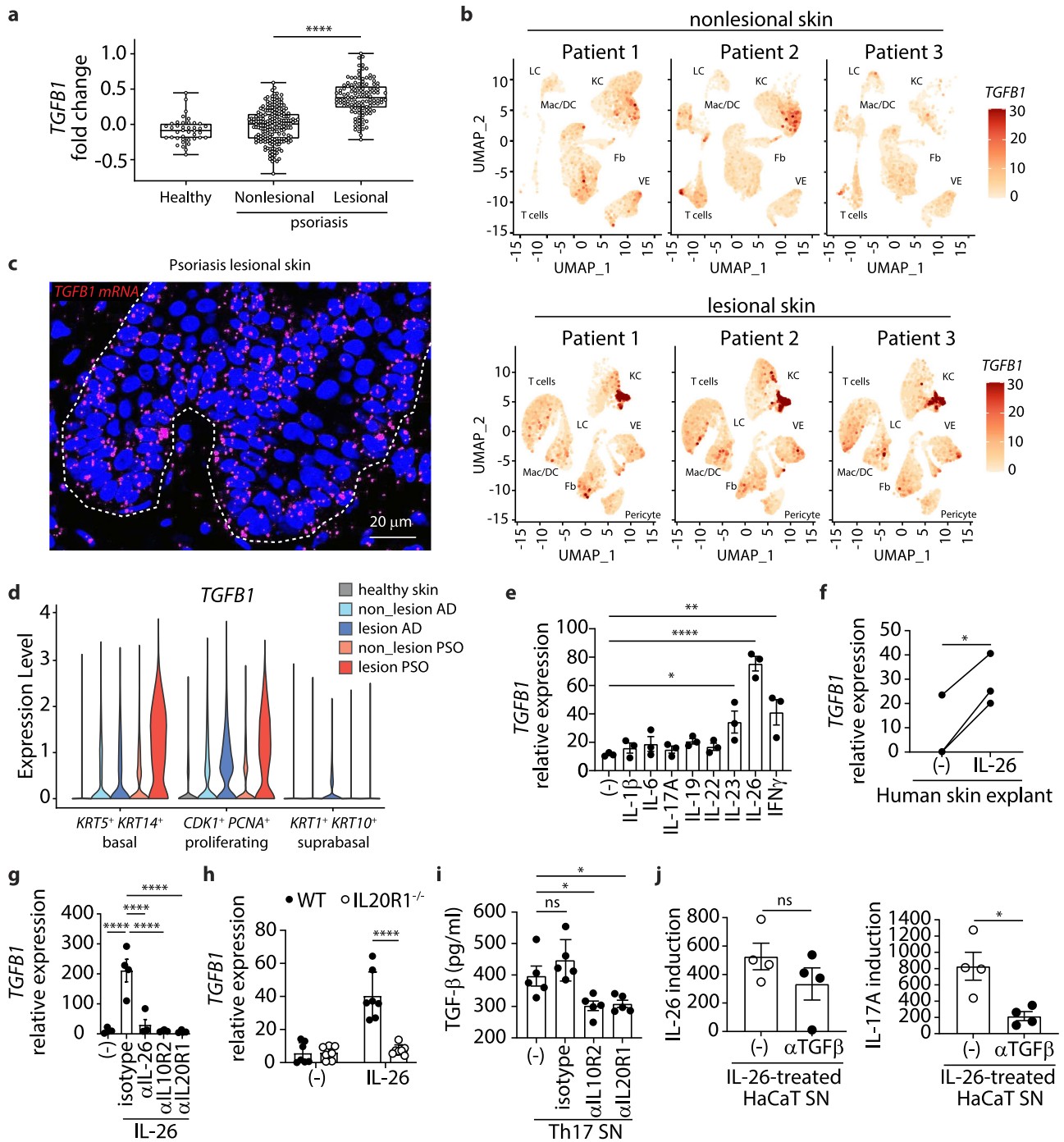

## Discussion

Our study identifies IL-26$^+$ T cells as an intermediate stage of $T_H$17 cell differentiation that is abundantly present in the blood circulation. These early IL-26$^+$ $T_H$17 cells infiltrate tissues, induce TGF-β1 expression, and mature into IL-17A-producing cells. By using a combination of scRNAseq and spatial transcriptomics we show that this process occurs in psoriatic skin lesions. Here, TGF-β1, specifically induced in basal keratinocytes by infiltrating IL-26$^+$ cells, drives $T_H$17 cell imprinting with conversion of IL-26$^+$ T cells into IL-17A producers. Thus, this study provides a unique mechanism by which the pathogenic $T_H$17 cell responses, associated with production of IL-17A, are controlled by an epithelial crosstalk involving IL-26.

The differentiation of naïve T cells into early $T_H$17 cell intermediates expressing IL-26 occurs rapidly in the presence of $T_H$17 polarizing cytokines and does not require TGF-β1. Subsequent

exposure to TGF-β1 drives the maturation of the early IL-26$^+$ $T_H$17 cell intermediates into IL-17A-single producing $T_H$17 cells, involving an IL-26$^+$ IL-17A$^+$ double-positive transition stage. Given that TGF-β upregulates RORγT[27], this transcription factor may drive the conversion of IL-26$^+$ $T_H$17 cell intermediates into IL-17A producers. However, whether the initial TGF-β-independent generation of IL-26$^+$ $T_H$17 cell intermediates involves the same transcription factor remains unclear. We have previously shown that IL-26$^+$ $T_H$17 cells are associated with an NF-κB signature and that IL-26 production is efficiently blocked by NF-κB inhibitors[28], suggesting a distinct transcription pathway for $T_H$17 cell intermediates.

Circulating IL-26$^+$ $T_H$17 cells are characterized by co-expression of the IL-1R[28]. These IL-1R$^+$ IL-26$^+$ T cells were previously shown to express regulatory T cell markers FOXP3, LRRC32, CTLA4 and CD25[28], suggesting a potential regulatory function of these cells. Interestingly, a

**Fig. 6 | Increased TGF-β1 expression by psoriatic keratinocytes induced by IL-26. a** Expression fold change of *TGFB1* in healthy skin (*n* = 40) and nonlesional (*n* = 199) and lesional (*n* = 120) skin from psoriasis patients using the SSF Bioinformatics hub. Data are represented as a box plot which bounds extends from the 25th to 75th percentiles, a middle line is plotted at the median, and whiskers go from the minimal to maximal value Data were statistically analyzed using one-way ANOVA followed by Tukey's multiple comparisons test. ****$p < 0.0001$. **b** UMAP projection of the single-cell transcriptomes of total cells from nonlesional (top) and lesional (bottom) skin of 3 psoriasis patients colored according to the expression level of *TGFB1*. LC Langherans cells, KC keratinocytes, Mac/DC macrophages and dendritic cells, Fb fibroblasts, VE vascular endothelial cells. **c** Confocal microscopy image of lesional psoriatic skin representative of 5 patients stained for TGF-β mRNA. Dashed line delineates the dermo-epidermal junction. **d** Violin plots of *TGFB1* expression by basal (*KRT5*+, *KRT14*+), proliferating (*CDK1*+, *PCNA*+), and differentiated suprabasal (*KRT1*+, *KRT10*+) keratinocytes using single-cell RNAseq data from nonlesional and lesional skin of atopic dermatitis and psoriasis patients. **e**, Expression of *TGFB1* in NHEK cells treated with different cytokines for 16 h. Data represent mean ± SEM of 3 biological replicates from one representative donor. Data were statistically analyzed using one-way ANOVA followed by Tukey's multiple comparisons test. *$p = 0.036$; **$p = 0.0063$, ****$p < 0.0001$. **f** Expression of *TGFB1* in healthy skin treated with IL-26 for 4 h. Data represent 3 independent donors. Data were statistically analyzed using two-tailed paired Student's *t*-test. *$p = 0.0122$. **g** Expression of *TGFB1* in NHEK cells treated overnight with IL-26 in the presence of blocking antibodies against IL-26, IL10R2, and IL20R1. Data represent mean ± SEM of 4 biological replicates from one donor. Data were statistically analyzed using one-way ANOVA followed by Tukey's multiple comparisons test. ****$p < 0.0001$. **h** Expression of *TGFB1* in WT or IL20R1$^{-/-}$ HaCaT cells treated overnight with IL-26. Data represent mean ± SEM of 7 biological replicates. Data were statistically analyzed using one-way ANOVA followed by Tukey's multiple comparisons test. ****$p < 0.0001$. **i**, Amounts of TGF-β1 produced by HaCaT cells treated overnight with T$_H$17 cell supernatants (from 5 independent donors) in the presence of blocking antibodies against IL10R2 and IL20R1, or control isotype. Data represent mean ± SEM. Data were statistically analyzed using one-way ANOVA followed by Tukey's multiple comparisons test. *$p = 0.0124$ (aIL10R2), *$p = 0.018$ (aIL20R1). **j** Amounts of IL-26 (left) and IL-17A (right) produced by blood T$_H$17 cells (from 4 independent donors) treated for 7 days with IL-26-treated HaCaT cell supernatants in the presence of blocking antibodies against TGF-β. Data represent mean ± SEM. Data were statistically analyzed using two-tailed unpaired Student's *t*-test. *$p = 0.014$. **e–h** Expression values measured by RT-qPCR were normalized to the reference gene GAPDH. Source data are provided as a Source Data file.

subpopulation of FOXP3$^+$ regulatory T cells expressing the IL-1R was found to be an early stage in T$_H$17 development[29,30], further supporting the possibility that IL-26$^+$ T$_H$17 cell intermediates are related to regulatory T cells. However, analysis of scRNAseq data from both blood and psoriatic skin lesions did not reveal T reg marker expression in IL-26$^+$ T cells, although we cannot exclude a dropout phenomenon. The possibility that these IL-26$^+$ T$_H$17 intermediates may be related to regulatory T cells and can switch their phenotype into pathogenic IL-17A-producing T$_H$17 cells upon skin infiltration remains intriguing and will have to be dissected in future studies.

IL-26 is implicated in the pathophysiology of several T$_H$17-driven autoimmune diseases including psoriasis[22], inflammatory bowel disease[31] and rheumatoid arthritis[32]. Our study unravels the link between IL-26 and pathogenic IL-17A responses by showing that IL-26 drives the maturation of T$_H$17 cell intermediates into IL-17A producers by inducing keratinocyte activation to produce TGF-β1. In psoriasis, keratinocyte activation occurs via the IL-26-receptor, a heterodimer composed of IL-10R1 and IL-20R1, which is widely expressed on epithelial cells. A similar process may occur in other barrier organs including the lungs, where a link between TGF-β and exaggerated T$_H$17 responses has been described[33]. Whether IL-26 also controls IL-17A expression in non-epithelial tissues remains to be determined. One possibility is that this may occur in IL-26R-independent, DNA-dependent manner through activation of immune cells via intracellular receptors such as TLR9 or cGAS-STING[5,6].

IL-26 is the key driver of TGF-β expression in psoriatic keratinocytes. In fact, IL-26 was the strongest TGF-β1 inducer in keratinocytes and its expression in psoriatic skin was closer to TGF-β1 than other psoriatic cytokines. The IL-26-mediated induction of TGF-β1 appears to be a general principle, but it preferentially occurs in psoriasis and not AD, despite the presence of similar numbers of IL-26$^+$ T$_H$17 cell intermediates in the dermis. The explanation for this phenomenon relies on the fact that only psoriatic T cells migrate into the epidermis, where they recognize epidermal and melanocytic autoantigens such as ADAMTSL5[34], LL37[35], whereas T cells in AD remain in the upper dermis, where they recognize antigens presented by dermal dendritic cells[36].

The vast majority of IL-26$^+$ T$_H$17 cell intermediates in healthy skin, nonlesional and lesional psoriatic skin express the marker CD103 suggesting that IL-26$^+$ T$_H$17 cell intermediates are tissue-resident and not directly derived from infiltrating circulating cells. This is line with recent reports showing that TRMs are major contributors to IL-26 production in inflammatory bowel disease[31,37,38]. Our findings raise the intriguing possibility that pathogenic T cells persist in psoriatic skin at sites of prior activation as TRMs, revert to an IL-26$^+$

state in the absence of TCR stimulation, and subsequently contribute to disease exacerbation through the mechanisms described in this study.

In summary, our data show that T$_H$17 cells control their own differentiation into mature IL-17A producing T$_H$17 cells through the secretion of IL-26 and induction of TGF-β1 in epithelial tissues. This site-dependent paracrine loop between T$_H$17 cells and epithelial cells may represent a mechanism to restrict IL17A expression to barrier tissues to avoid unfocused systemic production of T$_H$17 cytokines. Our finding also suggests that blocking IL-26 may be a beneficial approach to intervene in conditions which manifest exaggerated IL-17A responses in epithelial tissues including T$_H$17 diseases such as psoriasis.

## Methods

### Human samples and datasets

Studies were approved by the institutional review boards and the local ethics committee of the Lausanne University Hospital CHUV, Switzerland, in accordance with the Helsinki Declaration and were reviewed by the ethical committee board of the canton of Vaud, Switzerland (CER-VD 2020-02204). Fresh leftover surgical tissue from healthy skin and biobanked FFPE psoriatic skin tissue stored in the Swiss Biobanking Platform (SBP)-accredited Dermatology biobank were obtained from consented patients (Supplementary Table 1). PBMC were isolated from blood buffy coats of consented healthy donors and obtained from the Interregional Blood Transfusion Center, Bern, Switzerland (Supplementary Table 1).

Microarray transcriptional profiles from 134 psoriasis patients generated by Fyhrquist, et al.[39] and available via the Skin Science Foundation's bioinformatics hub [https://biohub.skinsciencefoundation.org] were selected and re-analyzed. Previously published single-cell RNA and TCR sequencing datasets were selected and re-analyzed for the current study. These datasets include data generated from 3 psoriasis patients by Reynolds et al.[24] and data from 10 psoriasis patients by Zhang et al.[25], deposited at ArrayExpress [www.ebi.ac.uk/arrayexpress/experiments/E-MTAB-8142] and at the European Genome-phenome Archive (EGA) [https://ega-archive.org], respectively (Supplementary Table 1).

### Blood T$_H$17 cell isolation, culture, and generation of T$_H$17 clones

T$_H$17 cells were isolated from healthy human peripheral blood mononuclear cells (PBMC) based on their CD4$^+$ CCR6$^+$CCR4$^+$ CXCR3$^-$ phenotype using the Easysep human T$_H$17 cell enrichment kit II (StemCell). In some experiments, purified T$_H$17 cells were cultured for 7 days in X vivo 15 serum-free media (Lonza) and activated by ImmunoCult™

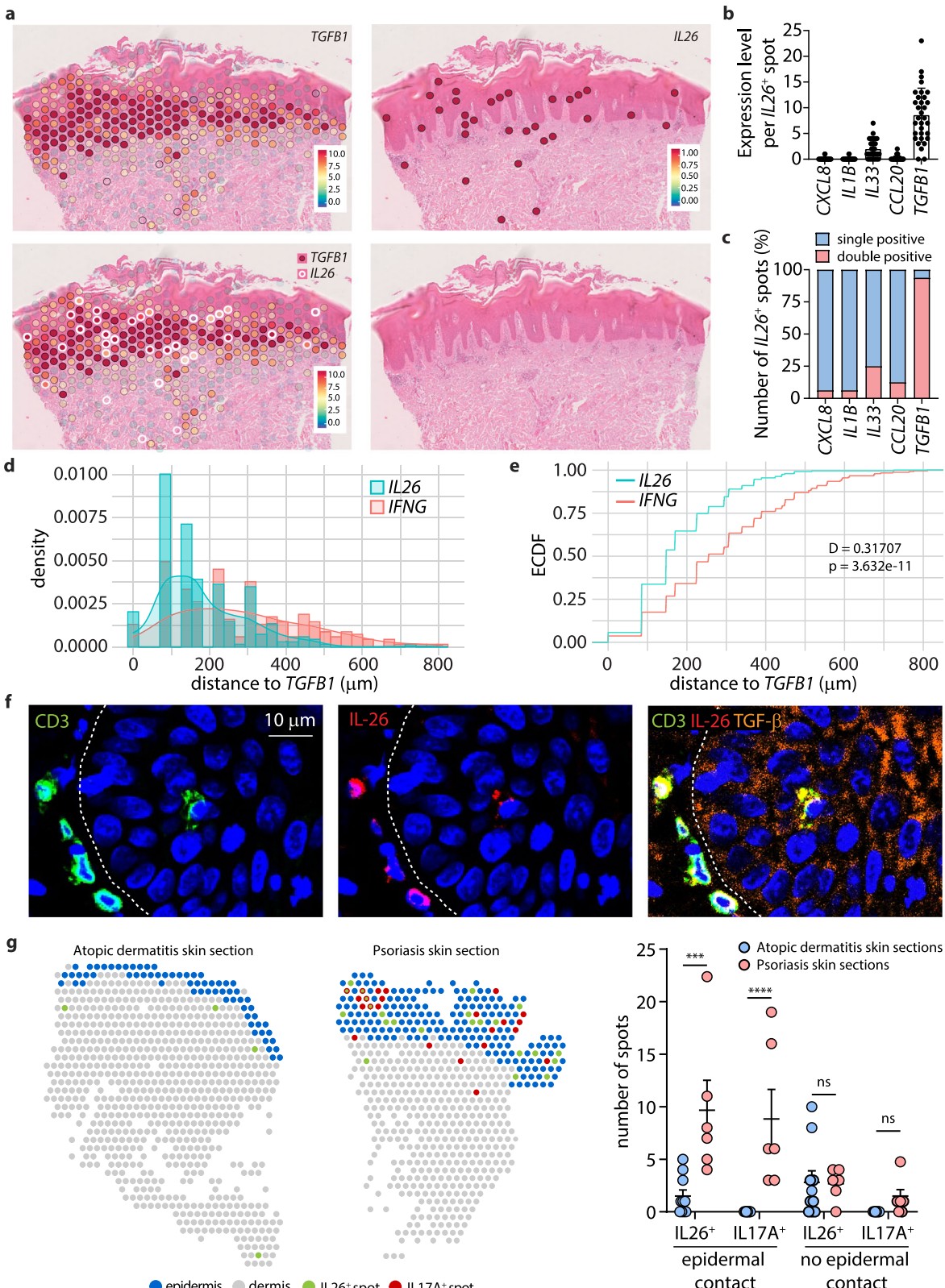

Human CD3/CD28/CD2 T Cell Activator (StemCell technologies) with or without TGF-β1 (10 ng/ml; StemCell). At the end of culture, cells were washed and either re-stimulated for 3 h with PMA and ionomycin using the Cell Stimulation Cocktail (eBioscience) and used for Flow-FISH analysis or re-stimulated for 24 h to generate supernatants. Cell-free supernatants were then harvested to measure IL-26 (Cusabio Biotech) and IL-17A (Mabtech) secretion by ELISA. In some

experiments, T$_H$17 cells were cultured for 5 days in X vivo 15 serum-free medium supplemented with 20% IL-26-treated HaCaT cell supernatant in the presence of anti-panTGFβ blocking antibody (50 µg/ml; Clone 1D11.16.8, BioXcell).

T$_H$17 cell clones were generated by limiting dilution in 96-well plates (0.8 cells per well) and expanded with allogeneic irradiated (45 Gy) PBMC (5×10$^4$ cells per well) in X vivo 15 serum-free media

**Fig. 7 | Spatial vicinity of IL26⁺ T cells with TGF-β1 expressing keratinocytes in psoriatic skin lesions. a** Spatial mapping of *TGFB1* and *IL26* expression in the lesional skin of a psoriasis patient. White circles show *IL26⁺TGFB1⁺* double positive spots. **b** Expression level of *CXCL8*, *IL1B*, *IL33*, and *CCL20* in IL26⁺ spots (*n* = 32) in one representative patient sample. Data represent mean ± SEM. **c** Percentages of IL26⁺ spots that are single-positive or double positive for *CXCL8*, *IL1B*, *IL33*, and *CCL20*. **d** Density distribution and **e** Empirical Cumulative Distribution Function (ECDF) of the shortest distance between spots containing IL26- or IFNG-positive T cells and spots containing TGFB1-expressing keratinocytes. The D and p values of two-sided two-sample Kolmogorov–Smirnov distance test are shown. **f** Confocal microscopy images of lesional psoriatic skin representative of 5 patients stained for CD3 (green), IL-26 (red), and TGF-β1 (orange). Dashed line delineates the dermo-epidermal junction. **g** Spatial mapping of IL26⁺ (green) and IL17A⁺ (red) spots in the lesional skin of an atopic dermatitis (left) and psoriasis (middle) patient. Spots over the dermis (gray) and epidermis (blue) are colored differently. Quantification of IL26⁺ and IL17A⁺ spots in contact or not with the epidermis in atopic dermatitis (*n* = 2 skin sections of 5 patients) and psoriasis (*n* = 2 skin sections of 3 patients) patient skin samples (right) is shown. Data are mean + SEM Data were statistically analyzed using two-way ANOVA followed by Sidak's multiple comparisons test. ***$p$ = 0.0006, ****$p$ < 0.0001. Source data are provided as a Source Data file.

(Lonza) containing antibiotics (1% Pen-Strep), IL-2 (100 U/ml) and phytohemagglutinin (PHA, 1 μg/ml; Life Technologies). Every five days, cells were split and fresh medium containing IL-2 and PHA was added until enough cells were obtained to perform experiments.

## In vitro T$_H$17 cell differentiation from naive T cells

Naive CD4 T cells were enriched from healthy donors PBMC with the Easysep Human Naive CD4 T cell isolation kit II (StemCell) and then sorted by flow cytometry as CD25⁻ CD8⁻ CD11c⁻ CD19⁻ CD56⁻ CD16⁻ CD14⁻ CD45RO⁻ CD4⁺ CD45RA⁺ cells to reach a purity > to 98% using a FACS Aria II device (BD Biosciences). Purified naive CD4 T cells were then cultured for 1–7 days in 96-weel U bottom plates at a density of 5 × 10⁴ cells per well in X vivo 15 serum-free media (Lonza) and activated by ImmunoCult™ Human CD3/CD28/CD2 T Cell Activator (StemCell technologies) in the presence of the following T$_H$17 polarizing cytokines: IL-1β (10 ng/ml, R&D Systems), IL-6 (20 ng/ml, R&D Systems), IL-23 (100 ng/ml, Peprotech), with or without TGFβ−1 (10 ng/ml, StemCell). To inhibit Th2 and Th1 polarization, anti-IL-4 and anti-IFNγ antibodies (10 μg/ml, eBioscience) were added into the cultures.

## Fluorescent in situ hybridization (FISH) analysis by flow cytometry (Flow-FISH)

In vitro differentiated and purified blood memory T$_H$17 cells were restimulated during 3 h with PMA and ionomycin using the Cell Stimulation Cocktail (eBioscience) and then stained with Alexa647-IL26, Alexa488-IL17A, Alexa568-IL22 and Alexa750-IFNγ probes using PrimeFlow RNA Assay (eBioscience) according to the manufacturer's protocol. In some experiments, intracellular cytokine staining of re-stimulated T$_H$17 cells and blocked for secretion by Brefledin A after 1 h was performed using anti-human IL-9 PeCy7 (BioLegend, 1/20), IL-10 PE (BD Biosciences, 1/20), IL-13 BV421 (BioLegend, 1/20) and IL-21 PE (BioLegend, 1/20) antibodies before proceeding with the hybridization step of the Prime-Flow Assay. Cells were acquired on a FACS LSR II SORP (BD Biosciences) and analyzed with FlowJo_v10.7.1. A detailed gating strategy for this analysis is shown in Supplementary Fig. 8.

## Isolation and culture of healthy skin

Several 6-mm punch biopsies were performed ex-vivo on residual healthy human skin obtained from excisional surgery. Biopsies were then rinsed with PBS containing antibiotics (1% Pen-Strep) and cultured in 48-well plates in the presence of 200 μl of DMEM 10% fetal bovine serum (FBS). Recombinant human.IL-26 (20 μg/ml) (R&D Systems, rhIL-26 monomer #1375-IL) was added to cultures for 6 h before harvesting. Harvested material was then homogenized in Trizol and followed by RNA extraction and real-time qPCR analysis. The cytokine concentration to use on healthy skin was determined based on a titration of *Escherichia coli*-derived recombinant human monomeric IL-26 for its ability to induce significant TGF-β1 production in skin biopsies, which provided an average EC50 value of 16 μg/ml.

## Immunofluorescence analysis

FFPE skin blocks were cut into 6 μm sections and placed on slides. Sections were first deparaffinized and rehydrated, then Heat-Induced Epitope Retrieval (HIER) was performed and sections were permeabilized with PBS 0.01% Triton. Samples were stained for 2 h at room temperature with the following primary antibodies: goat anti-human IL-26/AK155 (R&D systems, #AF1375, 1/100), mouse anti-human TGF beta-1 (Thermo Fisher Scientific, # MA5-16949, 1/500), rabbit anti-human CD3 (Ventana, # 790-4341, ready to use). Sections were then stained for 30 min at room temperature with the following fluorescently-labeled secondary antibodies: donkey anti-rabbit IgG (H + L) Alexa Fluor 488 (Invitrogen, #A-32790, 1/500), chicken anti-mouse IgG (H + L) Alexa Fluor 647 (Invitrogen, #A-21463, 1/500), donkey anti-goat IgG (H + L) Alexa Fluor 546 (Invitrogen, #A-32790, 1/500). For RNA ISH, TGFB1 mRNA (#400881) was detected in skin using RNAScope® Multiplex Fluorescent V2 Assay following the manufacturer's instruction (Advanced Cell Diagnostics, Inc). Sections were then labeled with OPAL 650 (Akoya Biosciences, #OP-001005, 1/1500). All slides were mounted with ProLong Gold antifade mounting with DAPI (Thermo Fisher Scientific). Images were acquired with a Zeiss LSM 700 confocal microscope and analyzed with Zen 2010 software.

## Keratinocyte stimulation

Normal human epidermal keratinocytes (NHEK, Sigma Aldrich) from 2 to 10 passages or the transformed keratinocyte cell line HaCaT (AddexBio) up to 15 passages were used. NHEK were cultured in Keratinocyte serum-free medium supplemented with human recombinant epidermal growth factor (rEGF) and bovine pituitary extract (BPE) (Gibco, ThermoFisher) and HaCaT cells in DMEM (Invitrogen) containing antibiotics (1% Pen-Strep) and 10% FBS. Keratinocytes were stimulated for 24 h with 100 ng/ml of the following cytokines: IL-1β (R&D Systems), IL-6 (R&D Systems), IL-17A (Biolegend), IL-19 (R&D Systems), IL-22 (StemCell), IL-23 (Peprotech), IL-26 (R&D Systems, rhIL-26 monomer #1375-IL), or IFNγ (Biolegend), or with 20% T$_H$17 supernatant. Cell-free supernatants were harvested after 24 h to measure TGF-β1 by ELISA (R&D Systems), and cells were used for RNA extraction and RT-qPCR analysis. The cytokine concentration to use on keratinocytes was determined based on a titration of *E. coli*-derived recombinant human monomeric IL-26 for its ability to induce TGFB1 mRNA expression in NHEK which provided an EC50 value of 36 ng/ml.

In some experiments, keratinocytes were pre-treated with blocking anti-IL-26 (Clone 84), anti-IL-10R2 (Clone # 90220), anti-IL-20R1 (Clone # 173707) (30 μg/ml) antibodies (R&D Systems), or mouse IgG1 isotype control 1 h prior stimulation.

## Generation of IL-20R1-deficient keratinocytes

To generate keratinocytes that are deficient for IL-20R1, we transfected HaCaT cells with three different CRISPR/Cas9 ribonucleoproteins (RNPs) targeting the IL20R1 gene using the GeneArt CRISPR gRNA Design Tool from Thermo Fisher Scientific. Synthesis of three different guide RNAs (gRNA) was performed using probes designed by the Invitrogen™ TrueDesign™ Genome Editor as following: IL20R1-gRNA-C1 5′-AGACACGTTATACTTCAGAT-3′ PAM TGG targeting exon 4, IL20R1-gRNA-C2 5′-CACTCTTTACTGCGTACACG-3′ PAM TGG targeting exon 5, and IL20R1-gRNA-C3 5′-CGTGTGGTTGGTCACACACT-3′ PAM GGG targeting exon 5. The three gRNA were synthesized in vitro using the GeneArt Precision gRNA Synthesis Kit (Thermo Fisher) and were purified using the gRNA Clean Up Kit. Transfection of cells with

CRISPR/Cas9 RNP complexes were performed by mixing 1 µg of TrueCut™ Cas9 Protein v2 (Thermo Fisher) with 240 ng of gRNA in 1.5 µl of Lipofectamine™ CRISPRMAX™ Cas9 Transfection Reagent (Thermo Fisher) for 10 min at room temperature. As negative control, a non-targeting TrueGuide™ sgRNA Negative Control was used. RNPs were then added to $5.10^4$ HaCaT cells in a 24-well plate for 2 days before knockdown evaluation by western blot. Cells were further cultured by limiting dilution to generate IL20R1$^{-/-}$ HaCaT clones.

## Analysis of psoriatic skin single-cell RNAseq—data visualization and cell counting

Single-cell RNAseq data generated by Reynolds et al.[24] were re-analyzed using processed datasets from nonlesional and lesional skin of three psoriasis patients (ArrayExpress database under accession number E-MTAB-8142). The gene counts matrix was analyzed with the R package Seurat v4.0[40]. Cell barcodes from nonlesional and lesional psoriatic skin were first filtered by the subset function using metadata information Status_site= "Psoriasis_non_lesion" and "Psoriasis_lesion". To analyze specifically dermal CD4 T cells, cell barcodes were further filtered on Tissue_layer = "dermis" and Celltype = "Th". Then, the filtered count matrix was normalized and scaled using the Seurat functions NormalizeData and ScaleData with all genes. A non-linear dimensional reduction of the dataset was run to allow Uniform Manifold Approximation and Projection (UMAP) exploration of the data using the Seurat functions FindVariableFeatures (selection.method = "vst", nfeatures = 7500) and RunPCA followed by FindNeighbors (dims = 1:15), FindClusters (resolution = 0.5), and RunUMAP (dims = 1:15). The FeaturePlot function was used to generate UMAP plots colored according to the expression of *IL17A*, *IL26*, *TGFB1*, or the TGF-β signature genes (*NR4A1*, *HSPA1A*, *CTLA4*, *DUSP6*, *LITAF*, *KLRG1*, *JUNB*, *NR4A2*, *PDCD1*, *CISH*, *EOMES*, and *SOCS3*). Cells expressing specific genes were counted based on the number of cell barcodes that were positive for the corresponding genes. For analyzing single-cell TCR sequencing data, datasets generated by Zhang et al.[25] were used and merged with the first dataset. The do_SankeyPlot function was used to analyze the TCR clonotypes sharing between cells (Blanco-Carmona, E. Generating publication ready visualizations for Single Cell transcriptomics using SCpubr. bioRxiv (2022) https://doi.org/10.1101/2022.02.28.482303).

## Analysis of psoriatic skin single-cell RNAseq—Pseudotime

The previous filtered count matrix was analyzed with the R package Monocle 3[41]. The new monocle object was generated with the Monocle 3 function new_cell_data_set using the expression data, metadata, and gene metadata from the Seurat object. Processing of the count matrix was done with the Monocle 3 function preprocess_cds and reduce_dimension. Both the Seurat embeddings and UMAP coordinates of the Seurat object were transferred to the monocle object to facilitate data visualization by generating identical UMAP plots. The trajectory of gene expression changes was constructed by using the Monocle 3 function learn_graph and the pseudotime was calculated with the function order_cells by using CCR7$^{hi}$ cells (CCR7 > 2.5) as beginning roots. Dynamics of gene expression over pseudotime were visualized by generating plots with the Monocle 3 function plot_genes_in_pseudotime using trend_formula = "~ splines::ns(pseudotime, df = 4)".

## Psoriatic skin microarray analysis

To analyze the expression of *TGFB1, TGFB2*, and *TGFB3* in healthy, nonlesional, and lesional skin of psoriasis patients, we used the Skin Science Foundation (SSF) bioinformatics hub. The SSF conducted a landscape analysis of publicly available human derived "omics" datasets. Current landscape analysis results integrate microarray experiments and can be explored on the website: https://biohub.skinsciencefoundation.org/labkey/home/project-begin.view.

## Spatial transcriptomics

Formalin-fixed paraffin-embedded (FFPE) tissue block from the lesional skin of a psoriasis patient was used for generating spatial transcriptomics using the Visium Spatial for FFPE Gene Expression kit according to the manufacturer's instructions (10X Genomics, Pleasanton, CA, USA). Basically, 8-µm-thick skin sections were cut with a microtome and placed into the Capture Area within the fiducial frame of the Visium Spatial slide using a water bath. The Visium slides containing FFPE tissue sections were then deparaffinized and stained with Hematoxylin and Eosin (H&E). The stained slides were then coverslipped and imaged using the PANNORAMIC 250 Flash digital scanner (3DHISTECH Ltd). After the coverslip is removed, a decrosslinking step was performed. Pairs of probes targeting specific genes of the human whole transcriptome were then added to the skin section for hybridization followed by ligation. Ligated probe products were then released and bound to spatially barcoded oligonucleotides that are specific to each spot. Libraries were then generated from the probe products by incorporating unique i7 and i5 sample indexes using the Dual Index Plate TS Set A. Libraries that met QC minimal requirements were finally sequenced using Illumina® NovaSeq (read length 100 with -25,000 reads/spot) and the Spatial Barcodes were used to associate the reads back to the skin section image for spatial mapping of gene expression using the Space Ranger pipeline. The processed matrix was finally analyzed with the R package Seurat v4.0 to generate gene expression images. The distance analysis was performed with the R package SpatialExperiment v1.4. For each keratinocyte expressing TGF-β1 (KRTDAP$^+$ and TGFB1$^+$ spot), a minimum distance to a T cell expressing IL-26 or IFN-γ (CD3D$^+$, CD3E$^+$, CD3G$^+$ and IL26$^+$ or IFNG$^+$ spot) was calculated based on the Euclidean distance. The difference between the two distance distributions was compared using the Kolmogorov–Smirnov test with a $p$ value threshold of 0.05 for significance[42,43].

## Statistics and reproducibility

GraphPad Prism (version 9) and Microsoft Excel (v2208) were used to generate charts and perform statistical analysis. The different statistical tests used in this study are described in the corresponding figure legends. Each experiment was reproduced at least 3 times. Data points represent biological replicates or different donors/patients and are shown as mean ± SEM unless mentioned otherwise in the figure legends. No data were excluded from the analyses. The experiments were randomized by splitting blood or keratinocyte donor cells into random wells before adding stimuli/treatments. The investigators in charge of the readouts assessment were blinded as experimental samples (antibody-stained cells, cell supernatants, tissue sections) were prepared and coded by other investigators.

## Reporting summary

Further information on research design is available in the Nature Portfolio Reporting Summary linked to this article.

# Data availability

Microarray transcriptional profiles generated by Fyhrquist et al.[39] are available via the Skin Science Foundation's bioinformatics hub [https://biohub.skinsciencefoundation.org]. Previously published single-cell RNA and TCR sequencing datasets by Reynolds et al.[24] and Zhang et al.[25], are deposited at ArrayExpress [www.ebi.ac.uk/arrayexpress/experiments/E-MTAB-8142] and at the European Genome-phenome Archive (EGA) [https://ega-archive.org/datasets/EGAD00001010106], respectively. Spatial transcriptomic datasets generated by Schabitz et al.[36], (GSE206391) and by us (GSE173706) are deposited at GEO data repository. Source data are provided with this paper.

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

## Acknowledgements

We thank I. Surbeck and A. Joncic for technical assistance. This work was funded by the Swiss National Science Foundation (310030B_182834 and 310030_204835) to M.G., by the German Research Foundation under Germany's Excellence Strategy (EXC 2155 "RESIST"—Project ID 390874280) to T.W., by the National Institute of Health (R01-AR080594) to R.L.M. and (P30-AR075043) to J.E.G.

## Author contributions

A.F. and F.S. conducted the in vitro experiments and associated analysis. J.D.D. performed the analysis of single-cell and spatial transcriptomics data with help from I.D., N.F., A.C.G, and R.G. F.K. and C.C. selected patients and provided skin samples. M. H., F.M., J.E.G., L.R, Y.L., T.W. and R.L.M. provided transcriptomics data. M.G. conceived and supervised the work, and wrote the manuscript along with J.D.D. and comments from co-authors.

## Competing interests

J.D.D. and M.G. are inventors of a patent entitled "IL-26 inhibitors" (WO2017009392A1). R.G. declares ownership in Ozette Technologies. The remaining authors declare no competing interests.
