## [Peer Review File · Nature Communications]

Differentiation of IL26+ TH17 intermediates into IL-17A producers via epithelial crosstalk in psoriasisREVIEWER COMMENTS

Reviewer #1 (Remarks to the Author):

It is noteworthy that the authors identify Th17 precursors in blood and describe how these cells differentiate into full pathogenic IL-17+ positive cells under the influence of TGFb within psoriatic tissue. In particular, the pseudotime and spatial experiments shown in Figures 4-7 are remarkably original, convincing, and exciting. A tour de force. This work advances our understanding of the immunology of psoriasis on a very detailed cellular level, and opens up new possibilities of targeting pathogenic T cells in this disease (e.g., IL-26 blockade, disruption of epithelial crosstalk).

I have a few suggestions for improvement for the authors to consider:

1) Why is this occurring in psoriasis? What is the "defect" in psoriatics that may trigger this process to occur in the first place? More specifically, does the blood of psoriatics differ compared to non-psoriatics regarding either 1) the number of IL-26+ and IL-17+ cells and/or 2) the ability of precursors to differentiate into IL-17+ cells? How would the blood cells of patients with atopic dermatitis (a non-Th17-mediated disease) compare to blood cells from psoriatics and healthy volunteers?

2) Similarly, are the tissue findings specific to psoriasis, or are they occurring in other inflammatory skin diseases? It would be nice to have at least one control inflammatory skin disease, such as atopic dermatitis, to show that the crosstalk and IL-17 cell localization is not present or different in some way.

MINOR COMMENTS:

1) I suggest a different title (word count pending) that incorporates the word "psoriasis," such as "TGFb1-dependent differentiation of IL-26+ Th17 cells into IL-17A producers in psoriasis."

2) I always prefer use of Oxford commas when listing items as I believe it improves clarity (pending journal style).

3) It should not be "an epithelial crosstalk," but rather just "epithelial crosstalk" (i.e., omit the "an" in all instances).

Reviewer #2 (Remarks to the Author):

Friess et al. identified IL-26 as factor defining immature Th17 cells in healthy blood. By recruitment into the skin, IL-26 triggers TGF-b production in keratinocytes that in turn leads to the terminal differentiation of IL-26+ immature Th17 cells into mature IL-17 producers. This is an interesting loop that brings together knowledge in the field and might explain the overrepresentation of Th17 cell sin psoriasis. The manuscript is well written and clearly presented. Please find my specific comment below.

Major comments

The manuscript is missing data on the frequency of IL-26+ cell skin Psoriasis patients compared to healthy controls. It is well appreciated that the frequency of IL-17+ T cell is higher in blood of patients and it would be of interest to understand if psoriasis patients per se do have a higher frequency of terminally differentiated Th17 cells or if the frequency of pre-differentiated cells that do need a further TGF-b hit is comparable between patients and controls.

The authors have nicely described the differentiation loop of IL-26 and TGF- β . However, it would be interesting to get a bit more information on the terminally differentiated Th17 cells. Which cytokines do they co-express? Other members of the IL-17 family and cytokines determining a pro- or anti-inflammatory function of Th17 cells would be of interest.

Th17 cells were predefined by expression of CCR4 and CCR6. However, this marker combination does not embrace all IL-17 producing CD4⁺ T cells. What is the overall frequency of IL-26⁺ T cells in blood of controls and psoriasis patients? Which cytokines do they co-produce? Is it possible to convert IL-26⁺ CCR4⁻CCR6⁻ T cells into IL-17 producers after stimulation with TGF β or is this an exclusive effect on CCR4⁺CCR6⁺ cells?

General comments

When reading through the introduction one might get the impression that it is unknown that Th17 cells are capable of IL-26 production and this is a novel finding in the submitted article. However, this is not the case as the association of IL-26 and Th17 cells is well known and needs to be included at least with a short note in the introduction

Line 76: "Because antibodies for intracellular IL-26 protein staining are not available". This is not entirely true as antibodies for e.g. intracellular flow cytometry can be purchased. If they are working or not is another matter of debate.

Line 84-86: "All clones contained high numbers of IL-26⁺IL17A⁻ T cells that were significantly more abundant than IL-17A⁺ T cells, similar to the observation made in primary TH17 cells (Fig. 1C and Fig.1D)." How many clones were obtained? What was the frequency of IL-26⁺ clones? Did the IL-26⁺ IL-17A⁻ clones produce other members of the IL-17 family? Fig. 1C/D just summarizes data from the selected four clones, however, it would be of interest to understand the overall presence of IL-26 in the generated T cell clones.

Fig. 2 D/E clearly shows that IL-26 expression is reduced by addition of TGF- β , however, this effect is not mentioned in the results so far. How do the authors explain the contrary effect of TGF- β on memory Th17 cells stimulated with TGF- β in Fig. 3?

Minor comments

Fig. 3 A: There seems to be a compensation artefact leading to an unspecific staining in the IL-26⁺IL-17⁺ quadrant. The reason for this artefact should be identified and eliminated.

The common way of writing genes and proteins should be followed throughout the manuscript (genes in italics without hyphen (IL-26) and proteins without italics and hyphen (IL-26))

Reviewer #3 (Remarks to the Author):

The authors demonstrate first that IL-26, a cytokine co-expressed by Th17 cells, is already expressed early in intermediates and/or precursors of Th17 cells that do not yet express IL-17A in the blood. By analyzing the requirements and kinetics for IL17A mRNA expression during in vitro Th17 differentiation they identify TGF- β 1 as a key cytokine of this induction process. They further demonstrate that IL-26 is one of the cytokines able to induce TGF- β 1 in keratinocytes/keratinocyte cell line or human skin explants, which then can induce IL-17A in these "IL-26⁺ Th17 intermediates" indicating a causal link. Specificity for TGF β 1 induction by IL26 in the HaCaT keratinocyte cell line is demonstrated by IL26 receptor subunit

blocking or IL20R1 knock-out HaCaT cells. Furthermore, supernatant from HaCaT keratinocytes can induce IL17A in TH17 intermediates in a TGFb1-dependent manner. This indicates that infiltrating IL26+ Th17 intermediates, which do not yet fully express IL17A, enter the psoriatic skin and become full-blown Th17 cells via a IL26/TGFB1 feed-forward loop.

Having demonstrated various aspects mechanistically using differentiated Th17 cells, they then evaluate whether this process is indeed operational in psoriatic skin. Using approaches such as spatial transcriptomics they show that IL26 and TGFb1 expression are present near each other supporting the hypothesis that local induction of TGFb1 in the psoriatic skin may be mediated by IL26. Furthermore, by single cell analysis of psoriatic dermal CD4 T cells, they show that a (partial; selected genes?) TGFb1 gene signature is highly enriched in IL17+ over IL26+ T-cells, which is in line with the postulated hypothesis of TGFb1 being a one of the key responsible cytokine for IL17A induction in these Th17 precursor/intermediate cells. Overall, these findings are of high interest to the scientific community as they shed a different light on TGF-b1 beyond its known role as a Th17 differentiation factor in psoriatic disease and how IL26, which has for long always been associated with Th17 cells, and TGFb1 are involved in this process. This is particularly interesting since the role of IL-26 (due to its absence in rodents) in inflammatory disease is still insufficiently understood.

Overall, the manuscript is well written but some of the statements made regarding with respect to the human psoriasis are not entirely supported by the human disease data. Their data provides a strong hypothesis but does not provide final proof.

In the discussion the authors also claim that “Our data show that TRMs residing in both healthy and non-lesional (pre-psoriatic) skin are poised to produce high levels of IL-26...” (line 282-283) yet the only data that could support this is shown in Figures 4D and 4E, where CCR7+ (more a marker of recirculation) CD4 T cells are located in the same location as IL-26+ T cells, which would be consistent with them being early intermediate Th17 cells. In contrast, CCR7- CD4 T cells (potential TRM cells) do not express IL26, but rather express IL17 which does not directly show that they are Trms. Do the authors also have single-cell data for CD103 or other markers of Trms to substantiate this point that they could show similarly than shown in Fig 4E?

A key aspect of the manuscript is the induction of TGF-b1 from keratinocytes by IL-26. In Figure 6D and 6G, they show that TGFb1 mRNA is most strongly induced by IL-26 in NHEK and HaCaT cells, which contrasts with Sa et al (Journal of Immunology, 2007, 178: 2229) where they mention that IL-26 was not able stimulate to stimulate NHEK proliferation (in line with lack of IL-20R1 expression). We do have similar experience. As far as HaCaT cells are concerned, earlier passages and length of cultivation significantly appear to affect the ability to observe for example IL-8 production induced by IL-26 (J Biol Chem 2004; 279:33343). These discrepancies raise the question whether culture conditions are responsible for the different results.

- Can the authors comment and if important provide more details in the materials and methods section about the culture conditions that were used and how they may influence e.g. NHEK differentiation state (are these cells more basal-like?)?

- Manuscript would benefit also from data on IL20R1 and IL10R2 expression for these in vitro experiments (NHEK and HaCaT) as well as the spatial transcriptomics data of lesional psoriatic skin.

IL-17A+ arising from IL26+ cells in psoriasis: In Figure 5, using TCR clonotypes, the authors shows that indeed IL17A+ can arise from IL26+ using a pseudo time analysis of single-cell data in psoriasis. However, they also show that IL17+ positive cells can arise from cells that are IL26- at early stages. It appears therefore that both is happening in psoriasis. Therefore, the authors discuss the results more comprehensively and reword lines 160-168 where this could be misunderstood as if all IL17+ Th17 cells derive from IL-26+ intermediates.

General formal aspects:

- In the text and figures also “Th17 intermediate cells” are called “Th17 cells”. This is

confusing and the term “Th17 cell” should only be used when they actually produce IL-17, otherwise use the term “Th17 intermediates” as in the abstract

- Nomenclature. Suggest to consistently use established nomenclature for protein (e.g. IL-17A, TGF- β 1) and RNA (e.g. IL17A, TGFB1) to ensure the reader understands whether he is looking at protein or RNA data.

- Whenever authors refer to TGF-b1, they should be precise and not use TGF-b and TGF-b1 interchangeably.

Comments to Figures:

Figure 2

- Figure 2C shows only 4 donors, whereas Figure 2B shows 5 donors. Is there one missing?

- Figure 2E shows only 3 donors, whereas Figure 2B shows 5 donors. Is there one missing?

Figure 3

- Legend indicates that blood memory Th17 cells (or intermediates) are restimulated with TGF-b

o Not clear how “memory” Th17 cells were selected. Are these indeed memory T-cells?

o How were they re-stimulated? Please mention in legend.

- Figure 3B shows n=3, whereas Figure 3C shows data is n=4. Are these different experiments?

- In Figure 3B at day7 for IL26-IL17+ cells there is no error bar – why?

- Authors may also want to consider using different symbols for the different populations for easier interpretation

- Figure 3C is missing the statistics

Figure 4

- Figure 4A: Do the authors also see TGF- β 1 receptor chain expression (TGFBRI and II) in dermal CD4 T cells (RNA and/or protein)? This would support the claim that dermal CD4 T cells can respond to TGF- β 1.

- Figure 4B: Unit of “Expression Level”? how normalized (size of biopsy or other)?

- Figure 4C: Number of TH17 or intermediate cells per what? (how were cells numbers normalized – surface area?)

- Figure 4E: CCR7 is not a bona-fide Trm marker. Consider marker such as CD103 as Trm marker

- Figure 4G: What is the Y-axis unit? If possible, include TGFB1 and R2 as well – one would expect those to be high early and then either decline or stay high? -> could provide information of skin T cells are only transiently or chronically sensitive to TGFb1 stimulation.

Figure 5

- Figure 5B: What is the Y-axis unit?

Figure 6

- It would be of importance to also demonstrate IL-20R1 and R2 expression by NHEK and HaCaT cells since others (e.g., Sa et al) were not able to activate these cells (at all or consistently dependent).

- Instead of “overnight” please add actual hours of incubation/culture

- Figure 6A: TGFB1 fold change, reference point – HV? Is the Y-axis log10?

- Figure 6D: TGFB1 fold change, reference point? Discuss in text also that IL-23 (not IL-23A) and IFN-gamma induce some TGFB1 mRNA, but IL-26 does so most strongly. All tested cytokines were used at 100ng/ml concentration – this could lead to differences in the observed TGFb1 response. A full titration curve and determination of EC50 levels would have been preferable since different cytokines have different potencies. Has this been done?

- Figure 6E: Were indeed 20 microgram/ml of IL-26 (Monomer or dimer; please provide R&D catalogue number) used? Why were so high levels used? Did you also measure TGF-b1 protein levels in the skin explant culture medium?

- Figure 6F: please provide mAb clone numbers and isotypes of antibodies used. What isotype control antibody was used?

- Figure 6I: Add “SN” to IL-26-treated HaCaT below the figures (missing)

Figure 7

- Figure 7A: To make the link between IL-26 and TGFb1 stronger it would be of value to also highlight those cells which express IL20R1 and IL20R2 near IL26. If data is available also include IL26 receptor subunits to Fig 7D and 7E.
- As a formal aspect– consider using similar labelling as for IL26 (wider line width, different colored ring) also for TGFb1 to allow better appreciation where markers are expressed and located

-

Gaps:

- IL-26 receptor expression (RNA and/or protein) not shown in primary keratinocytes, HaCaT cells and in psoriatic lesions e.g., by spatial transcriptomics
- Co-location of TGFb1 close to IL17A+ Th17 cells in psoriatic lesions not demonstrated
- Are psoriatic CD4 T cells showing the TGFb1 gene signature indeed in Th17 cells? (e.g. expressing RORGT and other hallmark genes)
- Many Th17 cells also do express IL17F. It would be of high interest to know if the postulated IL26/TGFb1 loop is also implicated in the expression of this cytokine.

Finally:

- Can authors speculate if the IL-26/TGFb1 driven “maturation” in skin operates also in Th2 cells which also express IL-26 (relevant for Atopic dermatitis; G. Reynolds et al., Science 371, eaba6500 (2021). DOI: 10.1126/science.aba6500)?

Reviewer #1:

It is noteworthy that the authors identify Th17 precursors in blood and describe how these cells differentiate into full pathogenic IL-17+ positive cells under the influence of TGF β within psoriatic tissue. In particular, the pseudotime and spatial experiments shown in Figures 4-7 are remarkably original, convincing, and exciting. A tour de force. This work advances our understanding of the immunology of psoriasis on a very detailed cellular level, and opens up new possibilities of targeting pathogenic T cells in this disease (e.g., IL-26 blockade, disruption of epithelial crosstalk).

We thank Reviewer #1 for his comments and for finding our work remarkably original, convincing and exciting.

I have a few suggestions for improvement for the authors to consider:

MAJOR COMMENTS:

- 1) *Why is this occurring in psoriasis? What is the "defect" in psoriatics that may trigger this process to occur in the first place? More specifically, does the blood of psoriatics differ compared to non-psoriatics regarding either the number of IL-26+ and IL-17+ cells and/or the ability of precursors to differentiate into IL-17+ cells? How would the blood cells of patients with atopic dermatitis (a non-Th17-mediated disease) compare to blood cells from psoriatics and healthy volunteers?*
- 2) *Similarly, are the tissue findings specific to psoriasis, or are they occurring in other inflammatory skin diseases? It would be nice to have at least one control inflammatory skin disease, such as atopic dermatitis, to show that the crosstalk and IL-17 cell localization is not present or different in some way.*

We thank the referee to raising these questions. We have now performed scRNAseq analysis to compare blood and skin of psoriasis and atopic dermatitis. We found the following:

- *IL26 and IL17A are undetectable in peripheral blood of both psoriasis and AD patients (scRNAseq, not shown).*
- *Similar high numbers of IL26+ Th17 intermediates are found in psoriasis and AD skin (**New Figure 5F**)*
- *IL26+ Th17 intermediates in psoriasis and AD have similar abilities to differentiate into IL17A producers (**New Figure 5G-H**), but high numbers of IL17A+ cells are only found in psoriatic skin (scRNAseq data, (**New Figure 5G-H**))*
- *Using spatial transcriptomics we found that only IL26+ Th17 intermediates in psoriasis but not in AD are in close contact with the epidermis (**New Figure 7G**) and associated with increased numbers of IL17A+ T cells in the epidermis (**New Figure 7G**).*
- *Production of TGF β by keratinocytes is abundant in psoriasis but not AD (**New Figure 6D**).*

Based on these new findings we propose the model in which IL-26+ Th17 intermediates can differentiate into IL-17A+ cells via induction of TGF- β if they get in contact with keratinocytes. This occurs in psoriasis, where T cells migrate into the epidermis to recognize epidermal and melanocytic autoantigens such ADAMTSL5 or LL37. By contrast T cells in AD IL-26+ Th17 intermediates remain in the upper dermis, where they recognize antigens presented by dermal dendritic cells. These elements have been added to the **New Discussion**.

MINOR COMMENTS:

1) I suggest a different title (word count pending) that incorporates the word "psoriasis," such as "TGFb1-dependent differentiation of IL-26+ Th17 cells into IL-17A producers in psoriasis."

We have changed the title to "Differentiation of IL26+ TH17 intermediates into IL-17A producers via epithelial crosstalk in psoriasis"

2) I always prefer use of Oxford commas when listing items as I believe it improves clarity (pending journal style).

We have added commas when listing items.

3) It should not be "an epithelial crosstalk," but rather just "epithelial crosstalk" (i.e., omit the "an" in all instances).

We have changed this.

Reviewer #2:

Friess et al. identified IL-26 as factor defining immature Th17 cells in healthy blood. By recruitment into the skin, IL-26 triggers TGF-b production in keratinocytes that in turn leads to the terminal differentiation of IL-26+ immature Th17 cells into mature IL-17 producers. This is an interesting loop that brings together knowledge in the field and might explain the overrepresentation of Th17 cell sin psoriasis. The manuscript is well written and clearly presented. Please find my specific comment below.

We thank the reviewer #2 for finding the described loop interesting and appreciating our manuscript as well written and clearly presented.

MAJOR COMMENTS:

1) The manuscript is missing data on the frequency of IL-26+ cell skin Psoriasis patients compared to healthy controls. It is well appreciated that the frequency of IL-17+ T cell is higher in blood of patients and it would be of interest to understand if psoriasis patients per se do have a higher frequency of terminally differentiated Th17 cells or if the frequency of pre-differentiated cells that do need a further TGF-b hit is comparable between patients and controls.

Please see comment 1 to reviewer 1. **New Fig. 5F** now shows the frequency of both IL26+ intermediates and IL17A+ T cells in healthy donors, as well as lesional and non-lesional psoriatic skin and atopic dermatitis. Healthy skin as well as non-lesional psoriatic and AD skin contain similar numbers of IL26+ intermediates, which are increased in both psoriasis and AD lesions. IL17A+ T cells are only present in psoriasis lesions, due to the fact that high numbers of IL26+Th17 intermediates get into contact with the epidermis in psoriasis but not in AD. Although we cannot rule out that some pre-differentiated IL17A+ T cells in psoriatic blood infiltrate skin lesion, our data suggests that the majority of IL17A+ cells in psoriatic skin derive from IL-26+ intermediates as they are transcriptionally homogenous (same U-map location, **Figure 4A**), 90% present a TGF-b signature (**Suppl Fig. 3C**), and they locate along the pseudotime trajectory originating from IL26+ intermediates (**Figure 4F**).

2) The authors have nicely described the differentiation loop of IL-26 and TGF-b. However, it would be interesting to get a bit more information on the terminally differentiated Th17 cells. Which

cytokines do they co-express? Other members of the IL-17 family and cytokines determining a pro-or anti-inflammatory function of Th17 cells would be of interest.

In Suppl Fig. 2 we show a comprehensive cytokine expression profile of IL26+ Th17 intermediates (depicted at early pseudotime) and fully differentiated IL17A+ Th17 (depicted at late pseudotime). Early IL-26+ Th17 intermediates appear to express *IL26* only. At intermediate stages of differentiation these cells co-express high levels of *IL22*, and start to co-express other cytokines including *GMCSF*, *TNF*, *IFNG*, *IL21*, *IL13* along with *IL17A*, *IL17F*. Differentiated IL17A+ cells at late pseudotime express high levels of *IL17A*, *IL17F*, *IFNG*, *GMCSF*, *TNF* and some residual *IL26*.

3) *Th17 cells were predefined by expression of CCR4 and CCR6. However, this marker combination does not embrace all IL-17 producing CD4+ T cells. What is the overall frequency of IL-26+ T cells in blood of controls and psoriasis patients? Which cytokines do they co-produce? Is it possible to convert IL-26+ CCR4-CCR6- T cells into IL-17 producers after stimulation with TGFb or is this an exclusive effect on CCR4+CCR6+ cells?*

To address this question we analyzed our psoriasis skin scRNAseq database, which unlike blood scRNAseq contain detectable levels of IL26+ T cells. We found that IL26+ cells that differentiate into IL17A+ cells consist of 70% CCR4+/CCR6+ and 30% of cells CCR4-/CCR6- cells (**P2P Fig.1**). Analyzing these IL26+ cells by pseudotime revealed that bona-fide IL26+ Th17 cells depicted at early pseudotime are 100% CCR4+/CCR6+, whereas negative cells appear later in the differentiation process (intermediate and late pseudotime) (**P2P Fig.1**). Together these data indicate that IL26+ Th17 intermediates are CCR4 and CCR6 positive and that expression is lost during the differentiation process into IL-17A+ cells.

Psoriasis dermal IL26+ Th17 intermediates differentiating into IL17A+ Th17 cells

P2P Figure 1. Both CCR6+/CCR4+ and CCR4-/CCR6- IL26-expressing cells can differentiate into Th17 cells.
A, Sankey diagram of the differentiation fate of CCR6+/CCR4+ and CCR6-/CCR4- IL26-expressing CD4 T cells into IL17A+ cells.

General comments

4) *When reading through the introduction one might get the impression that it is unknown that Th17 cells are capable of IL-26 production and this is a novel finding in the submitted article. However, this is not the case as the association of IL-26 and Th17 cells is well known and needs to be included at least with a short note in the introduction.* We have corrected this by mentioning "IL-26 is another cytokine produced by pathogenic TH17 cells" in the Introduction.

- 5) **Line 76: “Because antibodies for intracellular IL-26 protein staining are not available”. This is not entirely true as antibodies for e.g. intracellular flow cytometry can be purchased. If they are working or not is another matter of debate.** We have rephrased with “Because we were unable to obtain reproducible data on intracellular IL-26 protein staining”.

- 6) **Line 84-86: “All clones contained high numbers of IL-26+IL17A- T cells that were significantly more abundant than IL-17A + T cells, similar to the observation made in primary TH17 cells (Fig. 1C and Fig.1D).” How many clones were obtained? What was the frequency of IL-26+ clones? Did the L-26+ IL-17A- clones produce other members of the IL-17 family? Fig. 1C/D just summarizes data from the selected four clones, however, it would be of interest to understand the overall presence of IL-26 in the generated T cell clones.** Around 700 Th17 single cells were plated for generating clones. We obtained 32 different clones that produced cytokines among which 21 were IL26 single positive and 11 were IL26/IL17A double positive. We did not obtain any IL17A single positive clones. This information was added to the results.

- 7) **Fig. 2 D/E clearly shows that IL-26 expression is reduced by addition of TGF- β , however, this effect is not mentioned in the results so far. How do the authors explain the contrary effect of TGF- β on memory Th17 cells stimulated with TGF- β in Fig. 3?** In Figure 3A the total cytokine-producing cells augment over time (and therefore also the IL26+ cells), however when we compare the percentages of IL26+ single positives, IL26+IL17A+ double positives and IL17A+ single positives during the culture period we see a clear decrease of IL26+ single positives (the IL26+ Th17 intermediates) and the subsequent appearance of IL17A+ single positive cells (quantified in **Figure 3B**).

MINOR COMMENTS:

- 8) **Fig. 3 A: There seems to be a compensation artefact leading to an unspecific staining in the IL-26+IL-17+ quadrant. The reason for this artefact should be identified and eliminated.** We tried new compensation calculations and gating strategies but were unable to eliminate these highly positive cells in the first quadrant of the figure. We therefore think that these are not due to a compensation artefact but is rather linked to the flow-FISH procedure.

- 9) **The common way of writing genes and proteins should be followed throughout the manuscript (genes in italics without hyphen (IL26) and proteins without italics and hyphen (IL-26)).** We have used the correct nomenclature throughout the manuscript to write genes and proteins, however we decided to maintain the expression of positive markers by cells in non-italicized without hyphen format (e.g. IL26+ T cells instead of *IL26+* T cells).

Reviewer #3

The authors demonstrate first that IL-26, a cytokine co-expressed by Th17 cells, is already expressed early in intermediates and/or precursors of Th17 cells that do not yet express IL-17A in the blood. By analyzing the requirements and kinetics for IL17A mRNA expression during in vitro Th17 differentiation they identify TGF-b1 as a key cytokine of this induction process. They further demonstrate that IL-26 is one of the cytokines able to induce TGF-b1 in keratinocytes/keratinocyte cell line or human skin explants, which then can induce IL-17A in these “IL-26+ Th17 intermediates” indicating a causal link. Specificity for TGFb1 induction by IL26 in the HaCaT keratinocyte cell line is demonstrated by IL26 receptor subunit blocking or IL20R1 knock-out HaCaT cells. Furthermore, supernatant from HaCaT keratinocytes can induce IL17A in TH17 intermediates in a TGFb1-dependent manner.

This indicates that infiltrating IL26+ Th17 intermediates, which do not yet fully express IL17A, enter the psoriatic skin and become full-blown Th17 cells via a IL26/TGFB1 feed-forward loop. Having demonstrated various aspects mechanistically using differentiated Th17 cells, they then evaluate whether this process is indeed operational in psoriatic skin. Using approaches such as spatial transcriptomics they show that IL26 and TGFb1 expression are present near each other supporting the hypothesis that local induction of TGFb1 in the psoriatic skin may be mediated by IL26. Furthermore, by single cell analysis of psoriatic dermal CD4 T cells, they show that a (partial; selected genes?) TGFb1 gene signature is highly enriched in IL17+ over IL26+ T-cells, which is in line with the postulated hypothesis of TGFb1 being a one of the key responsible cytokine for IL17A induction in these Th17 precursor/intermediate cells. Overall, these findings are of high interest to the scientific community as they shed a different light on TGF-b1 beyond its known role as a Th17 differentiation factor in psoriatic disease and how IL26, which has for long always been associated with Th17 cells, and TGFb1 are involved in this process. This is particularly interesting since the role of IL-26 (due to its absence in rodents) in inflammatory disease is still insufficiently understood. Overall, the manuscript is well written but some of the statements made regarding with respect to the human psoriasis are not entirely supported by the human disease data. Their data provides a strong hypothesis but does not provide final proof.

We thank Reviewer #3 for considering our findings of high interest to the scientific community and for stating that our manuscript is well written. We have now addressed the concerns raised:

- 1) In the discussion the authors also claim that “Our data show that TRMs residing in both healthy and non-lesional (pre-psoriatic) skin are poised to produce high levels of IL-26...” (line 282-283) yet the only data that could support this is shown in Figures 4D and 4E, where CRR7+ (more a marker of recirculation) CD4 T cells are located in the same location as IL-26+ T cells, which would be consistent with them being early intermediate Th17 cells. In contrast, CCR7- CD4 T cells (potential TRM cells) do not express IL26, but rather express IL17 which does not directly show that they are Trms. Do the authors also have single-cell data for CD103 or other markers of Trms to substantiate this point that they could show similarly than shown in Fig 4E? We now show CD103 (ITGAE) expression on IL-26+ intermediates in **New Suppl Fig 2C**. We found that most of IL-26+ Th17 intermediates in healthy skin, non-lesional and lesional psoriatic skin were CD103 positive consistent with their TRM nature. The CCR7 marker*
- 2) A key aspect of the manuscript is the induction of TGF-b1 from keratinocytes by IL-26. In Figure 6D and 6G, they show that TGFb1 mRNA is most strongly induced by IL-26 in NHEK and HaCaT cells, which contrasts with Sa et al (Journal of Immunology, 2007, 178: 2229)*

where they mention that IL-26 was not able stimulate to stimulate NHEK proliferation (in line with lack of IL-20R1 expression). We do have similar experience.

We disagree with the reviewer that keratinocytes do not express the IL-26 receptor. Hor et al. (J Biol Chem 2004; 279:33343) show IL-20R1 mRNA expression in HaCaT cells and the Sa et al. paper observed that *IL20R1* expression in normal keratinocytes varies from donor to donor. Our scRNAseq analysis shows that both healthy and psoriatic keratinocytes express *IL10RB* and *IL20RA* (**P2P Fig 2A**). Moreover, spatial RNAseq analysis shows that *IL20RA* expression is confined to the lower epidermis in psoriatic skin (**P2P Fig 2B**). Altogether, these data demonstrate that keratinocytes do express the two chains of the IL-26 receptor. However, in accordance to the reviewer's comment we do not observe keratinocyte proliferation upon IL-26 stimulation as reported by Sa et al. (**P2P Fig 2C**). This does not exclude the activation of the IL-26R, which was also confirmed by Hor et al, showing STAT3 phosphorylation in keratinocytes and IL-8 production.

A

B

C

P2P Figure 2. Keratinocytes express IL-26 receptor but do not proliferate in response to IL-26.

A, UMAP projection of the single cell transcriptomes of cells from the skin of 5 healthy donors (top) and 3 psoriasis patients (bottom) colored according to the expression level of *IL10RB* (left) and *IL20RA* (right).

B, Spatial mapping of *IL10RB* (left) and *IL20RA* (right) expression in the lesional skin of a psoriasis patient.

C, Kinetics of confluency of normal human epidermal keratinocytes (NHEK) cultured in the presence (blue) or not (red) of IL26. Data of each replicate (left) or the mean \pm SD (right) are shown.

3) As far as HaCaT cells are concerned, earlier passages and length of cultivation significantly appear to affect the ability to observe for example IL-8 production induced by IL-26 (J Biol Chem 2004; 279:33343). These discrepancies raise the question whether culture conditions are responsible for the different results. All the information regarding the culture conditions are in the material and methods section. HaCaT cells were purchased at AddexBio and subcultured for up to 15 passages.

4) Can the authors comment and if important provide more details in the materials and methods section about the culture conditions that were used and how they may influence e.g. NHEK differentiation state (are these cells more basal-like?)? We added the information that between 2 and 10 passages of the purchased NHEK were performed. Additional information regarding the generation of NHEK can be found on the provider website (Sigma).

5) Manuscript would benefit also from data on IL20R1 and IL10R2 expression for these in vitro experiments (NHEK and HaCaT) as well as the spatial transcriptomics data of lesional psoriatic skin. Our scRNAseq analysis shows that both healthy and psoriatic keratinocytes express *IL10RB* and *IL20RA* (P2P Fig 2A). Moreover, spatial RNAseq analysis shows that *IL20RA* expression is confined to the lower epidermis in psoriatic skin (P2P Fig 2B). Altogether, these data demonstrate that keratinocytes express the two chains of the IL-26 receptor.

6) IL-17A+ arising from IL26+ cells in psoriasis: In Figure 5, using TCR clonotypes, the authors shows that indeed IL17A+ can arise from IL26+ using a pseudo time analysis of single-cell data in psoriasis. However, they also show that IL17+ positive cells can arise from cells that are IL26- at early stages. It appears therefore that both is happening in psoriasis. Therefore, the authors discuss the results more comprehensively and reword lines 160-168 where this could be misunderstood as if all IL17+ Th17 cells derive from IL-26+ intermediates. We have rephrased this.

General formal aspects:

7) In the text and figures also "Th17 intermediate cells" are called "Th17 cells". This is confusing and the term "Th17 cell" should only be used when they actually produce IL-17, otherwise use the term "Th17 intermediates" as in the abstract. We have changed the text accordingly

8) Nomenclature. Suggest to consistently use established nomenclature for protein (e.g. IL-17A, TGF-β1) and RNA (e.g. IL17A, TGFB1) to ensure the reader understands whether he is looking at protein or RNA data. We have changed the text accordingly.

9) Whenever authors refer to TGF-b1, they should be precise and not use TGF-b and TGF-b1 interchangeably. We have changed the text accordingly.

Comments to Figures:

Figure 2

10) Figure 2C shows only 4 donors, whereas Figure 2B shows 5 donors. Is there one missing? No. these are distinct experiments with 4 and 5 donors, respectively.

11) Figure 2E shows only 3 donors, whereas Figure 2B shows 5 donors. Is there one missing? No, these timecourse experiments are distinct from experiments in 2B.

Figure 3

12) Legend indicates that blood memory Th17 cells (or intermediates) are restimulated with TGF- β

o **Not clear how “memory” Th17 cells were selected. Are these indeed memory T-cells?**

These are CCR4/CCR6+ blood Th17. We did rework and omit the word memory.

o **How were they re-stimulated?** Now mentioned in the legend.

13) Figure 3B shows n=3, whereas Figure 3C shows data is n=4. Are these different experiments?

Again, experiments for flow-FISH and ELISA are not the same experiments.

14) In Figure 3B at day 7 for IL26-IL17+ cells there is no error bar – why? The error bars are very small and do therefore not appear. We have uploaded the source data as supplementary info.

15) Authors may also want to consider using different symbols for the different populations for easier interpretation. Done.

16) Figure 3C is missing the statistics. Stats with significance have been added.

Figure 4

17) Figure 4A: Do the authors also see TGF- β 1 receptor chain expression (TGFBR1 and II) in dermal CD4 T cells (RNA and/or protein)? This would support the claim that dermal CD4 T cells can respond to TGF- β 1. Yes, IL-26+ Th17 intermediates express the receptor, which is downregulated on IL-17A+ cells consistent with their activation (**New Suppl. Fig 3A**)

18) Figure 4B: Unit of “Expression Level”? how normalized (size of biopsy or other?)? As described in the material and methods, single-cell RNAseq raw counts are analyzed using the R package Seurat v4.0, which employs a global-scaling normalization method “LogNormalize” that normalizes the feature expression measurements for each cell by the total expression, multiplies this by a scale factor (10,000 by default), and log-transforms the result.

19) Figure 4C: Number of TH17 or intermediate cells per what? (how were cells numbers normalized – surface area?) Number of Th17 cells/biopsy. Added additional information to legend.

20) Figure 4E: CCR7 is not a bona-fide Trm marker. Consider marker such as CD103 as Trm marker. CCR7 was used as an early resting T cell diff. marker to orient the pseudotime trajectory (similarly to Szabo *et al.* Nat Commun. 2019). We now show CD103 (*ITGAE*) expression on IL-26+ intermediates to define TRMs (**New Suppl Fig 2C**).

21) Figure 4G: What is the Y-axis unit? The x-axis is the inferred pseudotime values and the y-axis is the normalized expression level of the gene. As explained in the material and methods, dynamics of gene expression over pseudotime were visualized by generating plots with the Monocle 3 function `plot_genes_in_pseudotime` using `trend_formula = "~ splines::ns(pseudotime, df=4)"`.

22) If possible, include TGFBR1 and R2 as well – one would expect those to be high early and then either decline or stay high? -> could provide information of skin T cells are only transiently or chronically sensitive to TGF β 1 stimulation.

Yes, IL-26 intermediates have the receptor, downregulated on IL-17A producers, consistent with their activation (**New Suppl. Fig 3A**)

Figure 5

23) **Figure 5B: What is the Y-axis unit?** The x-axis is the inferred pseudotime values and the y-axis is the normalized expression level of the gene. As explained in the material and methods, dynamics of gene expression over pseudotime were visualized by generating plots with the Monocle 3 function `plot_genes_in_pseudotime` using `trend_formula = "~ splines::ns(pseudotime, df=4)"`.

Figure 6

24) **It would be of importance to also demonstrate IL-20R1 and R2 expression by NHEK and HaCaT cells since others (e.g., Sa et al) were not able to activate these cells (at all or consistently dependent).** Our scRNAseq analysis shows that both healthy and psoriatic keratinocytes express *IL10RB* and *IL20RA* (P2P Fig 2A). Moreover, spatial RNAseq analysis shows that *IL20RA* expression is confined to the lower epidermis in psoriatic skin (P2P Fig 2B). Altogether, these data demonstrate that keratinocytes express the two chains of the IL-26 receptor.

25) **Instead of "overnight" please add actual hours of incubation/culture.** 16h, added to the new version of the manuscript.

26) **Figure 6A: TGFB1 fold change, reference point – HV? Is the Y-axis log10?** In order to integrate all datasets selected for meta-analysis, the expression data were normalized using a modification of the RatioA: $Y_i = \log_2(X_i - \text{mean}(X_i \text{ control}))$. For each gene (Y_i), the mean expression across all healthy control samples was calculated ($\text{mean}(X_i \text{ control})$) for each experiment independently. This experiment-specific mean expression of control samples was then used as an experiment-specific normalization factor by subtracting it from every expression value (X_i) for that gene in every sample, including healthy, lesional and non-lesional, for that experiment. Log2 transformation of this ratio results in a distribution of normalized expression values for control samples centered around 0 in every experiment while retaining differential expression in test samples.

27) **Figure 6D: TGFB1 fold change, reference point?** Expression values measured by RT-qPCR were normalized to the reference gene GAPDH. Info added to figure legend

28) **Discuss in text also that IL-23 and IFN-gamma induce some TGFB1 mRNA, but IL-26 does so most strongly. All tested cytokines were used at 100ng/ml concentration – this could lead to differences in the observed TGFB1 response. A full titration curve and determination of EC50 levels would have been preferable since different cytokines have different potencies. Has this been done?** TGFB mRNA expression was measured by RT-qPCR in NHEK following a range of IL-26 concentrations. We calculated an EC50 of 35.6 ng/ml (P2P Figure 3A). Although low levels of TGFB1 can be induced in keratinocytes by IL-23 and IFN-gamma, we did not find vicinity of TGFB1 to these cytokines when compared to IL-26 in the spatial transcriptomics analysis. The high TGFB1 levels induced in vitro plus the vicinity in-situ indicate that IL-26 is the key TGF- β inducer in psoriatic keratinocytes. This has been added to the discussion.

P2P Figure 3. Dose response curve of IL-26 stimulation of keratinocytes.

A, Expression of TGFβ1 in NHEK stimulated with increasing concentrations of IL-26 assessed by RT-qPCR. Calculated EC₅₀ is shown.

B, Production of TGF-β by skin explants from 3 different donors stimulated with increasing concentrations of IL-26 assessed by ELISA. Calculated EC₅₀ is shown.

29) Figure 6E: Were indeed 20 microgram/ml of IL-26 (Monomer or dimer; please provide R&D catalogue number) used? Recombinant human IL-26 monomer Catalog #1375-IL. Information was added to the Mat&Meth section.

30) Why were so high levels used? Did you also measure TGF-β1 protein levels in the skin explant culture medium? We previously showed that the natural IL-26 is more potent than recombinant IL-26 due to differences in posttranscriptional modifications between natural and recombinant proteins including glycosylation (Meller et al. Nature Immunology, 2015). Moreover, it is difficult to recapitulate microconcentrations of IL-26 that may occur in inflamed tissues in ex vivo experiments. Therefore, high amounts of recombinant IL-26 are required to reach significant activity. Amounts of TGFβ produced were quantified in the skin explant supernatants from 3 different donors stimulated with increasing concentrations of IL-26. We found that at least 20 μg/ml of IL-26 were required to induce its production for 2 out of 3 donors (**P2P Figure 3B**).

31) Figure 6F: please provide mAb clone numbers and isotypes of antibodies used. What isotype control antibody was used? OK info added to Mat & Meth section.

32) Figure 6I: Add "SN" to IL-26-treated HaCaT below the figures (missing) Added HACAT SN to the Figure.

Figure 7

33) Figure 7A: To make the link between IL-26 and TGFβ1 stronger it would be of value to also highlight those cells, which express IL20R1 and IL20R2 near IL26. If data is available also include IL26 receptor subunits to Fig 7D and 7E. We think that the hypothesis that the high expression levels of the 2 subunits correlates with the TGF-β production in keratinocytes is not valid, as the receptor may be downregulated upon activation by IL-26.

34) As a formal aspect– consider using similar labelling as for IL26 (wider line width, different colored ring) also for TGFβ1 to allow better appreciation where markers are expressed and located. We have improved the Figure legend to make it clearer, as follows: White circles show IL26+TGFβ1+ double positive spots.

Gaps:

35) *IL-26 receptor expression (RNA and/or protein) not shown in primary keratinocytes, HaCaT cells and in psoriatic lesions e.g., by spatial transcriptomics.* Please see answer to comment #2.

36) *Co-location of TGF β 1 close to IL17A+ Th17 cells in psoriatic lesions not demonstrated.* By spatial transcriptomics we found that the majority of IL-17A+ are associated to TGF β 1 expression (P2P Figure 4A and Figure 7G).

A

B

C

P2P Figure 4. IL-26⁺ and IL-17A⁺ T cells localize close to TGF β ⁺ keratinocytes and express ROR γ t.

A, Spatial mapping of *IL26* (left) and *IL17A* (right) expression by T cells along with *TGF β 1* expression by keratinocytes in the lesional skin of a psoriasis patient.

B, Dynamics of ROR γ t gene expression in skin CD4 T cells over pseudotime. Solid lines show the expression average.

C, Expression of ROR γ t by IL26⁺, IL17A⁺, and IFNG⁺ T cells from lesional psoriatic skin. Data were statistically analyzed using one-way ANOVA followed by Tukey's multiple comparisons test. *****p*<0.0001.

37) *Are psoriatic CD4 T cells showing the TGF β 1 gene signature indeed in Th17 cells? (e.g. expressing ROR γ t and other hallmark genes)* Yes, both IL26 Th17 intermediates and differentiated IL-17A⁺ cells express similar levels of ROR γ t (*ROR*) (P2P Figure 4B-C). As control, IFNG⁺ cells express significantly lower levels of ROR γ t.

38) Many Th17 cells also do express IL17F. It would be of high interest to know if the postulated IL26/TGFb1 loop is also implicated in the expression of this cytokine. Yes differentiated IL-17A+ T cells co-express IL-17F (see **Suppl Fig 2D**).

Finally:

39) Can authors speculate if the IL-26/TGFb1 driven “maturation” in skin operates also in Th2 cells which also express IL-26 (relevant for Atopic dermatitis; G. Reynolds et al., Science 371, - eaba6500 (2021). DOI: 10.1126/science.aba6500)?

See answers to Reviewer 1&2.

REVIEWER COMMENTS

Reviewer #1 (Remarks to the Author):

Thank you for addressing my concerns. The AD data are interesting and add significantly to the paper.

Reviewer #3 (Remarks to the Author):

1) Do the authors also have single-cell data for CD103 or other markers of Trms to substantiate this point that they could show similarly than shown in Fig 4E? We now show CD103 (ITGAE) expression on IL-26+ intermediates in New Suppl Fig 2C. We found that most of IL-26+ Th17 intermediates in healthy skin, non-lesional and lesional psoriatic skin were CD103 positive consistent with their TRM nature.

Response: Thanks for incorporating this important information. Question resolved.

2) A key aspect of the manuscript is the induction of TGF- β 1 from keratinocytes by IL-26. In Figure 6D and 6G, they show that TGF β 1 mRNA is most strongly induced by IL-26 in NHEK and HaCaT cells, which contrasts with Sa et al (Journal of Immunology, 2007, 178: 2229) where they mention that IL-26 was not able stimulate to stimulate NHEK proliferation (in line with lack of IL-20R1 expression). We do have similar experience.

We disagree with the reviewer that keratinocytes do not express the IL-26 receptor. Hor et al. (J Biol Chem 2004; 279:33343) show IL-20R1 mRNA expression in HaCaT cells and the Sa et al. paper observed that IL20R1 expression in normal keratinocytes varies from donor to donor. Our scRNAseq analysis shows that both healthy and psoriatic keratinocytes express IL10RB and IL20RA (P2P Fig 2A). Moreover, spatial RNAseq analysis shows that IL20RA expression is confined to the lower epidermis in psoriatic skin (P2P Fig 2B). Altogether, these data demonstrate that keratinocytes do express the two chains of the IL-26 receptor. However, in accordance to the reviewer's comment we do not observe keratinocyte proliferation upon IL-26 stimulation as reported by Sa et al. (P2P Fig 2C). This does not exclude the activation of the IL-26R, which was also confirmed by Hor et al, showing STAT3 phosphorylation in keratinocytes and IL-8 production.

Response: Thanks for providing the data in P2P Fig2A-C. Since this is relevant information, it would be good to add P2P Fig 2A and 2B as a supplementary to the manuscript. Question resolved.

P2P Figure 2. Keratinocytes express IL-26 receptor but do not proliferate in response to IL-26.

A, UMAP projection of the single cell transcriptomes of cells from the skin of 5 healthy donors (top) and 3 psoriasis patients (bottom) colored according to the expression level of IL10RB (left) and IL20RA (right).

B, Spatial mapping of IL10RB (left) and IL20RA (right) expression in the lesional skin of a psoriasis patient.

C, Kinetics of confluency of normal human epidermal keratinocytes (NHEK) cultured in the presence (blue) or not (red) of IL26. Data of each replicate (left) or the mean \pm SD (right) are shown.

3) As far as HaCaT cells are concerned, earlier passages and length of cultivation significantly appear to affect the ability to observe for example IL-8 production induced by IL-26 (J Biol Chem 2004; 279:33343). These discrepancies raise the question whether culture

conditions are responsible for the different results. All the information regarding the culture conditions are in the material and methods section. HaCaT cells were purchased at AddexBio and subcultured for up to 15 passages.

Response: Thank you, for adding this information.

4) Can the authors comment and if important provide more details in the materials and methods section about the culture conditions that were used and how they may influence e.g. NHEK differentiation state (are these cells more basal-like?)? We added the information that between 2 and 10 passages of the purchased NHEK were performed. Additional information regarding the generation of NHEK can be found on the provider website (Sigma).
Response: Thank you, for adding this information.

4) Manuscript would benefit also from data on IL20R1 and IL10R2 expression for these in vitro experiments (NHEK and HaCaT) as well as the spatial transcriptomics data of lesional psoriatic skin. Our scRNAseq analysis shows that both healthy and psoriatic keratinocytes express IL10RB and IL20RA (P2P Fig 2A). Moreover, spatial RNAseq analysis shows that IL20RA expression is confined to the lower epidermis in psoriatic skin (P2P Fig 2B). Altogether, these data demonstrate that keratinocytes express the two chains of the IL-26 receptor.

Response: See comment above. Suggest adding this data to the supplementary information.

6) IL-17A+ arising from IL26+ cells in psoriasis: In Figure 5, using TCR clonotypes, the authors shows that indeed IL17A+ can arise from IL26+ using a pseudo time analysis of single-cell data in psoriasis. However, they also show that IL17+ positive cells can arise from cells that are IL26- at early stages. It appears therefore that both is happening in psoriasis. Therefore, the authors discuss the results more comprehensively and reword lines 160-168 where this could be misunderstood as if all IL17+ Th17 cells derive from IL-26+ intermediates. We have rephrased this. Response: Thank you.

General formal aspects:

7) In the text and figures also "Th17 intermediate cells" are called "Th17 cells". This is confusing and the term "Th17 cell" should only be used when they actually produce IL-17, otherwise use the term "Th17 intermediates" as in the abstract. We have changed the text accordingly.

Response: Thank you.

8) Nomenclature. Suggest to consistently use established nomenclature for protein (e.g. IL-17A, TGF- β 1) and RNA (e.g. IL17A, TGFB1) to ensure the reader understands whether he is looking at protein or RNA data. We have changed the text accordingly.

Response: Thank you.

9) Whenever authors refer to TGF-b1, they should be precise and not use TGF-b and TGF-b1 interchangeably. We have changed the text accordingly.

Response: Please double-check throughout manuscript. For example: It still says TGF-b although cells have been stimulated with TGF-b1.

Comments to Figures:

Figure 2

10) Figure 2C shows only 4 donors, whereas Figure 2B shows 5 donors. Is there one missing? No. these are distinct experiments with 4 and 5 donors, respectively. Thanks. Resolved.

11) Figure 2E shows only 3 donors, whereas Figure 2B shows 5 donors. Is there one missing? No, these timecourse experiments are distinct from experiments in 2B.

Response: Thanks. Resolved.

Figure 3

12) Legend indicates that blood memory Th17 cells (or intermediates) are restimulated with TGF- β

o Not clear how “memory” Th17 cells were selected. Are these indeed memory T-cells? These are CCR4/CCR6+ blood Th17. We did rework and omit the word memory.

Response: Thanks. Resolved.

o How were they re-stimulated? Now mentioned in the legend. Response: Thanks. Resolved.

13) Figure 3B shows n=3, whereas Figure 3C shows data is n=4. Are these different experiments? Again, experiments for flow-FISH and ELISA are not the same experiments.

Response: Thanks for the clarification. Resolved.

14) In Figure 3B at day 7 for IL26-IL17+ cells there is no error bar – why? The error bars are very small and do therefore not appear. We have uploaded the source data as supplementary info. Response: Thanks. Resolved.

15) Authors may also want to consider using different symbols for the different populations for easier interpretation. Done. Response: Thanks. Resolved.

16) Figure 3C is missing the statistics. Stats with significance have been added. Response: Thanks. Resolved.

Figure 4

17) Figure 4A: Do the authors also see TGF- β 1 receptor chain expression (TGFBRI and II) in dermal CD4 T cells (RNA and/or protein)? This would support the claim that dermal CD4 T cells can respond to TGF- β 1. Yes, IL-26+ Th17 intermediates express the receptor, which is downregulated on IL-17A+ cells consistent with their activation (New Suppl. Fig 3A).

Response: Thanks for including the data. Resolved.

18) Figure 4B: Unit of “Expression Level”? how normalized (size of biopsy or other)? As described in the material and methods, single-cell RNAseq raw counts are analyzed using the R package Seurat v4.0, which employs a global-scaling normalization method “LogNormalize” that normalizes the feature expression measurements for each cell by the total expression, multiplies this by a scale factor (10,000 by default), and log-transforms the result.

Response: Thanks for the clarification.

19) Figure 4C: Number of TH17 or intermediate cells per what? (how were cells numbers normalized – surface area?) Number of Th17 cells/biopsy. Added additional information to legend.

Response: Thanks for adding.

20) Figure 4E: CCR7 is not a bona-fide Trm marker. Consider marker such as CD103 as Trm marker. CCR7 was used as an early resting T cell diff. marker to orient the pseudotime trajectory (similarly to Szabo et al. Nat Commun. 2019). We now show CD103 (ITGAE) expression on IL-26+ intermediates to define TRMs (New Suppl Fig 2C).

Response: Thanks for including the data. Resolved.

21) Figure 4G: What is the Y-axis unit? The x-axis is the inferred pseudotime values and the y-axis is the normalized expression level of the gene. As explained in the material and methods, dynamics of gene expression over pseudotime were visualized by generating plots

with the Monocle 3 function `plot_genes_in_pseudotime` using `trend_formula = "~splines::ns(pseudotime, df=4)"`.

Response: Thanks for the clarification.

22) If possible, include TGFBR1 and R2 as well – one would expect those to be high early and then either decline or stay high? -> could provide information of skin T cells are only transiently or chronically sensitive to TGFb1 stimulation.

Yes, IL-26 intermediates have the receptor, downregulated on IL-17A producers, consistent with their activation (New Suppl. Fig 3A)

Response: Thanks for including the data. Resolved.

Figure 5

23) Figure 5B: What is the Y-axis unit? The x-axis is the inferred pseudotime values and the y-axis is the normalized expression level of the gene. As explained in the material and methods, dynamics of gene expression over pseudotime were visualized by generating plots with the Monocle 3 function `plot_genes_in_pseudotime` using `trend_formula = "~splines::ns(pseudotime, df=4)"`.

Response: Thanks for the clarification.

Figure 6

24) It would be of importance to also demonstrate IL-20R1 and R2 expression by NHEK and HaCaT cells since others (e.g., Sa et al) were not able to activate these cells (at all or consistently dependent). Our scRNAseq analysis shows that both healthy and psoriatic keratinocytes express IL10RB and IL20RA (P2P Fig 2A). Moreover, spatial RNAseq analysis shows that IL20RA expression is confined to the lower epidermis in psoriatic skin (P2P Fig 2B). Altogether, these data demonstrate that keratinocytes express the two chains of the IL-26 receptor.

Response: See comment above. Suggest adding this data to the supplementary information.

25) Instead of “overnight” please add actual hours of incubation/culture. 16h, added to the new version of the manuscript.

Response: Thank you.

26) Figure 6A: TGFb1 fold change, reference point – HV? Is the Y-axis log10? In order to integrate all datasets selected for meta-analysis, the expression data were normalized using a modification of the RatioA: $Y_i = \log_2(X_i - \text{mean}(X_i \text{ control}))$. For each gene (Y_i), the mean expression across all healthy control samples was calculated ($\text{mean}(X_i \text{ control})$) for each experiment independently. This experiment-specific mean expression of control samples was then used as an experiment-specific normalization factor by subtracting it from every expression value (X_i) for that gene in every sample, including healthy, lesional and non-lesional, for that experiment. Log2 transformation of this ratio results in a distribution of normalized expression values for control samples centered around 0 in every experiment while retaining differential expression in test samples.

Response: Thanks for the clarification.

27) Figure 6D: TGFb1 fold change, reference point? Expression values measured by RT-qPCR were normalized to the reference gene GAPDH. Info added to figure legend. Thank you.

28) Discuss in text also that IL-23 and IFN-gamma induce some TGFb1 mRNA, but IL-26 does so most strongly. All tested cytokines were used at 100ng/ml concentration – this could lead to differences in the observed TGFb1 response. A full titration curve and determination of EC50 levels would have been preferable since different cytokines have different potencies. Has this been done? TGFb mRNA expression was measured by RT-qPCR in NHEK following a range of IL-26 concentrations. We calculated an EC50 of 35.6 ng/ml (P2P

Figure 3A). Although low levels of TGFB1 can be induced in keratinocytes by IL-23 and IFN-gamma, we did not find vicinity of TGFB1 to these cytokines when compared to IL-26 in the spatial transcriptomics analysis. The high TGFB1 levels induced in vitro plus the vicinity in situ indicate that IL-26 is the key TGF- β inducer in psoriatic keratinocytes. This has been added to the discussion.

Response: Argument of using expression in combination with proximity in situ as a measure of potential importance in vivo accepted and now included in discussion. Resolved.

P2P Figure 3. Dose response curve of IL-26 stimulation of keratinocytes.

A, Expression of TGFB1 in NHEK stimulated with increasing concentrations of IL-26 assessed by RT-qPCR. Calculated EC50 is shown.

B, Production of TGF- β by skin explants from 3 different donors stimulated with increasing concentrations of IL-26 assessed by ELISA. Calculated EC50 is shown.

29) Figure 6E: Were indeed 20 microgram/ml of IL-26 (Monomer or dimer; please provide R&D catalogue number) used? Recombinant human IL-26 monomer Catalog #1375-IL. Information was added to the Mat&Meth section.

Response: Could not see this in the M&M section; maybe forgotten to add. Please check/add.

30) Why were so high levels used? Did you also measure TGF- β 1 protein levels in the skin explant culture medium? We previously showed that the natural IL-26 is more potent than recombinant IL-26 due to differences in posttranscriptional modifications between natural and recombinant proteins including glycosylation (Meller et al. Nature Immunology, 2015). Moreover, it is difficult to recapitulate microconcentrations of IL-26 that may occur in inflamed tissues in ex vivo experiments. Therefore, high amounts of recombinant IL-26 are required to reach significant activity. Amounts of TGF β produced were quantified in the skin explant supernatants from 3 different donors stimulated with increasing concentrations of IL-26. We found that at least 20 μ g/ml of IL-26 were required to induce its production for 2 out of 3 donors (P2P Figure 3B).

Response: It is understood that IL-26 is not an "easy" protein due to its multiple forms (monomer, dimer, complexes, charge distribution etc). Therefore, it is critical to clearly mention which IL-26 form of the protein has been used (R&D sells different ones) by providing the catalogue number. Authors may also want to consider adding in M&M that IL-26 used ("E.coli-derived, recombinant, monomeric IL-26") has been titrated on NHEK and provide EC50 value of 36 ng/ml). For the ex vivo skin experiments, the use of 20 μ g/ml IL-26 on skin is roughly 500x higher than the NHEK-derived concentration, but maybe warranted due to lower access/penetration of IL-26 into the tissue. Minor issue but important for those who would like to repeat the experiments.

31) Figure 6F: please provide mAb clone numbers and isotypes of antibodies used. What isotype control antibody was used? OK info added to Mat & Meth section. Thank you.

32) Figure 6I: Add "SN" to IL-26-treated HaCaT below the figures (missing) Added HACAT SN to the Figure.

Response: Thank you.

Figure 7

33) Figure 7A: To make the link between IL-26 and TGF β 1 stronger it would be of value to also highlight those cells, which express IL20R1 and IL20R2 near IL26. If data is available also include IL26 receptor subunits to Fig 7D and 7E. We think that the hypothesis that the high expression levels of the 2 subunits correlates with the TGF- β production in keratinocytes is not valid, as the receptor may be downregulated upon activation by IL-26.

Response: It was not the intention to assume a direct correlation of expression levels with TGF- β 1 production, but rather verify proximity of producer cells and responder cells more

closely via TGFβ receptor expression also in Psoriasis disease tissue. This may not be easily seen in the spatial transcriptomics data in case RNA is short-lived or their RNA is far downregulated.

34) As a formal aspect— consider using similar labelling as for IL26 (wider line width, different colored ring) also for TGFβ1 to allow better appreciation where markers are expressed and located. We have improved the Figure legend to make it clearer, as follows: White circles show IL26+TGFβ1+ double positive spots. Response: Thank you.

Gaps:

35) IL-26 receptor expression (RNA and/or protein) not shown in primary keratinocytes, HaCaT cells and in psoriatic lesions e.g., by spatial transcriptomics. Please see answer to comment #2.

Response: Adding P2P Fig2A and Fig 2B to supplementary would also answer/resolve this question.

36) Co-location of TGFβ1 close to IL17A+ Th17 cells in psoriatic lesions not demonstrated. By spatial transcriptomics we found that the majority of IL-17A+ are associated to TGFβ+ expression (P2P Figure 4A and Figure 7G).

Response: P2P Figure 4A shows this. Thank you. However, no TGFβ1 expression highlighted on top of IL17A expression in Figure 7G. Please check if this was missed when adding the new Figure 7G.

P2P Figure 4. IL-26+ and IL-17A+ T cells localize close to TGFβ+ keratinocytes and express RORγt.

A, Spatial mapping of IL26 (left) and IL17A (right) expression by T cells along with TGFβ1 expression by keratinocytes in the lesional skin of a psoriasis patient.

B, Dynamics of RORγt gene expression in skin CD4 T cells over pseudotime. Solid lines show the expression average.

C, Expression of RORC by IL26+, IL17A+, and IFNG+ T cells from lesional psoriatic skin. Data were statistically analyzed using one-way ANOVA followed by Tukey's multiple comparisons test. ****p<0.0001.

37) Are psoriatic CD4 T cells showing the TGFβ1 gene signature indeed in Th17 cells? (e.g. expressing RORγt and other hallmark genes) Yes, both IL26 Th17 intermediates and differentiated IL-17A+ cells express similar levels of RORγt (RORC) (P2P Figure 4B-C). As control, IFNG+ cells express significantly lower levels of RORγt .

Response: Thank you.

38) Many Th17 cells also do express IL17F. It would be of high interest to know if the postulated IL26/TGFβ1 loop is also implicated in the expression of this cytokine. Yes differentiated IL-17A+ T cells co-express IL-17F (see Suppl Fig 2D).

Response: Resolved. Thank you.

Finally:

39) Can authors speculate if the IL-26/TGFβ1 driven “maturation” in skin operates also in Th2 cells which also express IL-26 (relevant for Atopic dermatitis; G. Reynolds et al., Science 371,

See answers to Reviewer 1&2.

Response: Thanks for adding this important information. Resolved.

REVIEWER COMMENTS

Reviewer #1 (Remarks to the Author):

Thank you for addressing my concerns. The AD data are interesting and add significantly to the paper.

We thank the reviewer for this comment.

Reviewer #3 (Remarks to the Author):

1) Do the authors also have single-cell data for CD103 or other markers of Trms to substantiate this point that they could show similarly than shown in Fig 4E?

We now show CD103 (ITGAE) expression on IL-26+ intermediates in New Suppl Fig 2C. We found that most of IL-26+ Th17 intermediates in healthy skin, non-lesional and lesional psoriatic skin were CD103 positive consistent with their TRM nature.

Response: Thanks for incorporating this important information. Question resolved.

Thank you.

2) A key aspect of the manuscript is the induction of TGF- β 1 from keratinocytes by IL-26. In Figure 6D and 6G, they show that TGF β 1 mRNA is most strongly induced by IL-26 in NHEK and HaCaT cells, which contrasts with Sa et al (Journal of Immunology, 2007, 178: 2229) where they mention that IL-26 was not able stimulate to stimulate NHEK proliferation (in line with lack of IL-20R1 expression). We do have similar experience.

We disagree with the reviewer that keratinocytes do not express the IL-26 receptor. Hor et al. (J Biol Chem 2004; 279:33343) show IL-20R1 mRNA expression in HaCaT cells and the Sa et al. paper observed that IL20R1 expression in normal keratinocytes varies from donor to donor. Our scRNAseq analysis shows that both healthy and psoriatic keratinocytes express IL10RB and IL20RA (P2P Fig 2A). Moreover, spatial RNAseq analysis shows that IL20RA expression is confined to the lower epidermis in psoriatic skin (P2P Fig 2B). Altogether, these data demonstrate that keratinocytes do express the two chains of the IL-26 receptor. However, in accordance to the reviewer's comment we do not observe keratinocyte proliferation upon IL-26 stimulation as reported by Sa et al. (P2P Fig 2C). This does not exclude the activation of the IL-26R, which was also confirmed by Hor et al, showing STAT3 phosphorylation in keratinocytes and IL-8 production.

Response: Thanks for providing the data in P2P Fig2A-C. Since this is relevant information, it would be good to add P2P Fig 2A and 2B as a supplementary to the manuscript. Question resolved.

We have now included data from P2P Fig2 into new supplementary Figure 5

3) As far as HaCaT cells are concerned, earlier passages and length of cultivation significantly appear to affect the ability to observe for example IL-8 production induced by IL-26 (J Biol Chem 2004; 279:33343). These discrepancies raise the question whether culture conditions are responsible for the different results.

All the information regarding the culture conditions are in the material and methods section. HaCaT cells were purchased at AddexBio and subcultured for up to 15 passages.

Response: Thank you, for adding this information.

Thank you.

4) Can the authors comment and if important provide more details in the materials and methods

section about the culture conditions that were used and how they may influence e.g. NHEK differentiation state (are these cells more basal-like?)?

We added the information that between 2 and 10 passages of the purchased NHEK were performed. Additional information regarding the generation of NHEK can be found on the provider website (Sigma).

Response: Thank you, for adding this information.

Thank you.

4) Manuscript would benefit also from data on IL20R1 and IL10R2 expression for these in vitro experiments (NHEK and HaCaT) as well as the spatial transcriptomics data of lesional psoriatic skin.

Our scRNAseq analysis shows that both healthy and psoriatic keratinocytes express IL10RB and IL20RA (P2P Fig 2A). Moreover, spatial RNAseq analysis shows that IL20RA expression is confined to the lower epidermis in psoriatic skin (P2P Fig 2B). Altogether, these data demonstrate that keratinocytes express the two chains of the IL-26 receptor.

Response: See comment above. Suggest adding this data to the supplementary information.

We have now included data from P2P Fig2 into new supplementary Figure 5

6) IL-17A+ arising from IL26+ cells in psoriasis: In Figure 5, using TCR clonotypes, the authors shows that indeed IL17A+ can arise from IL26+ using a pseudo time analysis of single-cell data in psoriasis. However, they also show that IL17+ positive cells can arise from cells that are IL26- at early stages. It appears therefore that both is happening in psoriasis. Therefore, the authors discuss the results more comprehensively and reword lines 160-168 where this could be misunderstood as if all IL17+ Th17 cells derive from IL-26+ intermediates.

We have rephrased this.

Response: Thank you.

Thank you.

General formal aspects:

7) In the text and figures also "Th17 intermediate cells" are called "Th17 cells". This is confusing and the term "Th17 cell" should only be used when they actually produce IL-17, otherwise use the term "Th17 intermediates" as in the abstract.

We have changed the text accordingly.

Response: Thank you.

Thank you.

8) Nomenclature. Suggest to consistently use established nomenclature for protein (e.g. IL-17A, TGF-β1) and RNA (e.g. IL17A, TGFB1) to ensure the reader understands whether he is looking at protein or RNA data.

We have changed the text accordingly.

Response: Thank you.

Thank you.

9) Whenever authors refer to TGF-b1, they should be precise and not use TGF-b and TGF-b1 interchangeably.

We have changed the text accordingly.

Response: Please double-check throughout manuscript. For example: It still says TGF-b although cells have been stimulated with TGF-b1.

We thank the reviewer for noticing it. Indeed, we forgot to update the figure legends accordingly. We have now used TGF- β 1 throughout the manuscript unless one cannot identify the TGF- β isoform (e.g. in vivo signature or findings from published studies)

Comments to Figures:

Figure 2

10) Figure 2C shows only 4 donors, whereas Figure 2B shows 5 donors. Is there one missing?

No, these are distinct experiments with 4 and 5 donors, respectively.

Thanks. Resolved.

Thank you.

11) Figure 2E shows only 3 donors, whereas Figure 2B shows 5 donors. Is there one missing?

No, these timecourse experiments are distinct from experiments in 2B.

Response: Thanks. Resolved.

Thank you.

Figure 3

12) Legend indicates that blood memory Th17 cells (or intermediates) are restimulated with TGF- β o Not clear how "memory" Th17 cells were selected. Are these indeed memory T-cells?

These are CCR4/CCR6+ blood Th17. We did rework and omit the word memory.

Response: Thanks. Resolved.

Thank you.

o How were they re-stimulated?

Now mentioned in the legend.

Response: Thanks. Resolved.

Thank you.

13) Figure 3B shows n=3, whereas Figure 3C shows data is n=4. Are these different experiments?

Again, experiments for flow-FISH and ELISA are not the same experiments.

Response: Thanks for the clarification. Resolved.

Thank you.

14) In Figure 3B at day 7 for IL26-IL17+ cells there is no error bar – why?

The error bars are very small and do therefore not appear. We have uploaded the source data as supplementary info.

Response: Thanks. Resolved.

Thank you.

15) Authors may also want to consider using different symbols for the different populations for easier interpretation.

Done.

Response: Thanks. Resolved.

Thank you.

16) Figure 3C is missing the statistics.

Stats with significance have been added.

Response: Thanks. Resolved.

Thank you.

Figure 4

17) Figure 4A: Do the authors also see TGF- β 1 receptor chain expression (TGFBR1 and II) in dermal CD4 T cells (RNA and/or protein)? This would support the claim that dermal CD4 T cells can respond to TGF- β 1.

Yes, IL-26+ Th17 intermediates express the receptor, which is downregulated on IL-17A+ cells consistent with their activation (New Suppl. Fig 3A).

Response: Thanks for including the data. Resolved.

Thank you.

18) Figure 4B: Unit of "Expression Level"? how normalized (size of biopsy or other?)?

As described in the material and methods, single-cell RNAseq raw counts are analyzed using the R package Seurat v4.0, which employs a global-scaling normalization method "LogNormalize" that normalizes the feature expression measurements for each cell by the total expression, multiplies this by a scale factor (10,000 by default), and log-transforms the result.

Response: Thanks for the clarification.

Thank you.

19) Figure 4C: Number of TH17 or intermediate cells per what? (how were cells numbers normalized – surface area?) Number of Th17 cells/biopsy.

Added additional information to legend.

Response: Thanks for adding.

Thank you.

20) Figure 4E: CCR7 is not a bona-fide Trm marker. Consider marker such as CD103 as Trm marker. CCR7 was used as an early resting T cell diff. marker to orient the pseudotime trajectory (similarly to Szabo et al. Nat Commun. 2019).

We now show CD103 (ITGAE) expression on IL-26+ intermediates to define TRMs (New Suppl Fig 2C).

Response: Thanks for including the data. Resolved.

Thank you.

21) Figure 4G: What is the Y-axis unit?

The x-axis is the inferred pseudotime values and the y-axis is the normalized expression level of the gene. As explained in the material and methods, dynamics of gene expression over pseudotime were visualized by generating plots with the Monocle 3 function `plot_genes_in_pseudotime` using `trend_formula = "~ splines::ns(pseudotime, df=4)"`.

Response: Thanks for the clarification.

Thank you.

22) If possible, include TGFBR1 and R2 as well – one would expect those to be high early and then either decline or stay high? -> could provide information of skin T cells are only transiently or chronically sensitive to TGF β 1 stimulation.

Yes, IL-26 intermediates have the receptor, downregulated on IL-17A producers, consistent with their activation (New Suppl. Fig 3A)

Response: Thanks for including the data. Resolved.

Thank you.

Figure 5

23) Figure 5B: What is the Y-axis unit?

The x-axis is the inferred pseudotime values and the y-axis is the normalized expression level of the gene. As explained in the material and methods, dynamics of gene expression over pseudotime were visualized by generating plots with the Monocle 3 function `plot_genes_in_pseudotime` using `trend_formula = "~ splines::ns(pseudotime, df=4)"`.

Response: Thanks for the clarification.

Thank you.

Figure 6

24) It would be of importance to also demonstrate IL-20R1 and R2 expression by NHEK and HaCaT cells since others (e.g., Sa et al) were not able to activate these cells (at all or consistently dependent).

Our scRNAseq analysis shows that both healthy and psoriatic keratinocytes express IL10RB and IL20RA (P2P Fig 2A). Moreover, spatial RNAseq analysis shows that IL20RA expression is confined to the lower epidermis in psoriatic skin (P2P Fig 2B). Altogether, these data demonstrate that keratinocytes express the two chains of the IL-26 receptor.

Response: See comment above. Suggest adding this data to the supplementary information.

We have now included data from P2P Fig2 into new supplementary Figure 5

25) Instead of "overnight" please add actual hours of incubation/culture.

16h, added to the new version of the manuscript.

Response: Thank you.

Thank you.

26) Figure 6A: TGFB1 fold change, reference point – HV? Is the Y-axis log10?

In order to integrate all datasets selected for meta-analysis, the expression data were normalized using a modification of the RatioA: $Y_i = \log_2(X_i - \text{mean}(X_i \text{ control}))$. For each gene (Y_i), the mean expression across all healthy control samples was calculated ($\text{mean}(X_i \text{ control})$) for each experiment independently. This experiment-specific mean expression of control samples was then used as an experiment-specific normalization factor by subtracting it from every expression value (X_i) for that gene in every sample, including healthy, lesional and non-lesional, for that experiment. Log2 transformation of this ratio results in a distribution of normalized expression values for control samples centered around 0 in every experiment while retaining differential expression in test samples.

Response: Thanks for the clarification.

Thank you.

27) Figure 6D: TGFB1 fold change, reference point? Expression values measured by RT-qPCR were normalized to the reference gene GAPDH.

Info added to figure legend.

Thank you.

Thank you.

28) Discuss in text also that IL-23 and IFN-gamma induce some TGFB1 mRNA, but IL-26 does so

most strongly. All tested cytokines were used at 100ng/ml concentration – this could lead to differences in the observed TGFb1 response. A full titration curve and determination of EC50 levels would have been preferable since different cytokines have different potencies. Has this been done?

TGFb mRNA expression was measured by RT-qPCR in NHEK following a range of IL-26 concentrations. We calculated an EC50 of 35.6 ng/ml (P2P Figure 3A). Although low levels of TGFb1 can be induced in keratinocytes by IL-23 and IFN-gamma, we did not find vicinity of TGFb1 to these cytokines when compared to IL-26 in the spatial transcriptomics analysis. The high TGFb1 levels induced in vitro plus the vicinity in-situ indicate that IL-26 is the key TGF-b inducer in psoriatic keratinocytes. This has been added to the discussion.

Response: Argument of using expression in combination with proximity in situ as a measure of potential importance in vivo accepted and now included in discussion. Resolved.

Thank you.

29) Figure 6E: Were indeed 20 microgram/ml of IL-26 (Monomer or dimer; please provide R&D catalogue number) used?

Recombinant human IL-26 monomer Catalog #1375-IL. Information was added to the Mat&Meth section.

Response: Could not see this in the M&M section; maybe forgotten to add. Please check/add.

The info was indeed added to the M&M section in the “Keratinocyte stimulation” subsection but was absent from the “Isolation and culture of healthy skin” subsection. This has been corrected.

30) Why were so high levels used? Did you also measure TGF-b1 protein levels in the skin explant culture medium?

We previously showed that the natural IL-26 is more potent than recombinant IL-26 due to differences in posttranscriptional modifications between natural and recombinant proteins including glycosylation (Meller et al. Nature Immunology, 2015). Moreover, it is difficult to recapitulate microconcentrations of IL-26 that may occur in inflamed tissues in ex vivo experiments. Therefore, high amounts of recombinant IL-26 are required to reach significant activity. Amounts of TGFb produced were quantified in the skin explant supernatants from 3 different donors stimulated with increasing concentrations of IL-26. We found that at least 20 µg/ml of IL-26 were required to induce its production for 2 out of 3 donors (P2P Figure 3B).

Response: It is understood that IL-26 is not an “easy” protein due to its multiple forms (monomer, dimer, complexes, charge distribution etc). Therefore, it is critical to clearly mention which IL-26 form of the protein has been used (R&D sells different ones) by providing the catalogue number. Authors may also want to consider adding in M&M that IL-26 used (“E.coli-derived, recombinant, monomeric IL-26”) has been titrated on NHEK and provide EC50 value of 36 ng/ml). For the ex vivo skin experiments, the use of 20ug/ml IL-26 on skin is roughly 500x higher than the NHEK-derived concentration, but maybe warranted due to lower access/penetration of IL-26 into the tissue. Minor issue but important for those who would like to repeat the experiments.

We have now added to the M&M section in the “keratinocyte stimulation” subsection that the cytokine concentration to use on keratinocytes was determined based on a titration of E.coli-derived recombinant human monomeric IL-26 for its ability to induce TGFb1 mRNA expression in NHEK which provided an EC50 value of 36 ng/ml. We also added in the “Isolation and culture of healthy skin” subsection that the cytokine concentration to use on healthy skin was determined based on a titration of E.coli-derived recombinant human monomeric IL-26 for its ability to induce significant TGF-b1 production in skin biopsies, which provided an average EC50 value of 16 ug/ml.

31) Figure 6F: please provide mAb clone numbers and isotypes of antibodies used. What isotype control antibody was used?

OK info added to Mat & Meth section.

Thank you.

Thank you.

32) Figure 6I: Add "SN" to IL-26-treated HaCaT below the figures (missing)

Added HACAT SN to the Figure.

Response: Thank you.

Thank you.

Figure 7

33) Figure 7A: To make the link between IL-26 and TGFb1 stronger it would be of value to also highlight those cells, which express IL20R1 and IL20R2 near IL26. If data is available also include IL26 receptor subunits to Fig 7D and 7E.

We think that the hypothesis that the high expression levels of the 2 subunits correlates with the TGF-b production in keratinocytes is not valid, as the receptor may be downregulated upon activation by IL-26.

Response: It was not the intention to assume a direct correlation of expression levels with TGF-b1 production, but rather verify proximity of producer cells and responder cells more closely via TGFb receptor expression also in Pso disease tissue. This may not be easily seen in the spatial transcriptomics data in case RNA is short-lived or their RNA to far downregulated.

Indeed, downregulation of the receptor mRNA upon triggering by IL-26 may interfere with the spatial distance calculations between producer and responder cells.

34) As a formal aspect– consider using similar labelling as for IL26 (wider line width, different colored ring) also for TGFb1 to allow better appreciation where markers are expressed and located.

We have improved the Figure legend to make it clearer, as follows: White circles show IL26+TGFb1+ double positive spots.

Response: Thank you.

Thank you.

Gaps:

35) IL-26 receptor expression (RNA and/or protein) not shown in primary keratinocytes, HaCaT cells and in psoriatic lesions e.g., by spatial transcriptomics.

Please see answer to comment #2.

Response: Adding P2P Fig2A and Fig 2B to supplementary would also answer/resolve this question.

We have now included data from P2P Fig2 into new supplementary Figure 5

36) Co-location of TGFb1 close to IL17A+ Th17 cells in psoriatic lesions not demonstrated.

By spatial transcriptomics we found that the majority of IL-17A+ are associated to TGFb+ expression (P2P Figure 4A and Figure7G).

Response: P2P Figure 4A shows this. Thank you. However, no TGFb1 expression highlighted on top of IL17A expression in Figure 7G. Please check if this was missed when adding the new Figure 7G.

Indeed, P2P Figure 4A is sufficient to show the spatial co-location of IL17A and TGFb1. Figure 7G should not have been mentioned in our previous responses.

37) Are psoriatic CD4 T cells showing the TGFb1 gene signature indeed in Th17 cells? (e.g. expressing RORGT and other hallmark genes)

Yes, both IL26 Th17 intermediates and differentiated IL-17A+ cells express similar levels of ROR γ t (RORC) (P2P Figure 4B-C). As control, IFNG+ cells express significantly lower levels of ROR γ t .

Response: Thank you.

Thank you.

38) Many Th17 cells also do express IL17F. It would be of high interest to know if the postulated IL26/TGFb1 loop is also implicated in the expression of this cytokine.

Yes differentiated IL-17A+ T cells co-express IL-17F (see Suppl Fig 2D).

Response: Resolved. Thank you.

Thank you.

Finally:

39) Can authors speculate if the IL-26/TGFb1 driven "maturation" in skin operates also in Th2 cells which also express IL-26 (relevant for Atopic dermatitis; G. Reynolds et al., Science 371,

See answers to Reviewer 1&2.

Response: Thanks for adding this important information. Resolved.

Thank you.

REVIEWERS' COMMENTS

Reviewer #3 (Remarks to the Author):

Thanks for addressing the remaining points. Manuscript will add significantly to our understanding of Th17 development and role of IL-26 locally in psoriatic lesional tissue.

REVIEWERS' COMMENTS

Reviewer #3 (Remarks to the Author):

Thanks for addressing the remaining points. Manuscript will add significantly to our understanding of Th17 development and role of IL-26 locally in psoriatic lesional tissue.

We thank the reviewer for this comment.